# COPILOT4D: LEARNING UNSUPERVISED WORLD MODELS FOR AUTONOMOUS DRIVING VIA DISCRETE DIFFUSION

**Lunjun Zhang**  **Yuwen Xiong**  **Ze Yang**  **Sergio Casas**  **Rui Hu**  **Raquel Urtasun**

Waabi    University of Toronto
{lzhang, yxiong, zyang, sergio, rhu, urtasun}@waabi.ai

## ABSTRACT

Learning world models can teach an agent how the world works in an unsupervised manner. Even though it can be viewed as a special case of sequence modeling, progress for scaling world models on robotic applications such as autonomous driving has been somewhat less rapid than scaling language models with Generative Pre-trained Transformers (GPT). We identify two reasons as major bottlenecks: dealing with complex and unstructured observation space, and having a scalable generative model. Consequently, we propose **Copilot4D**, a novel world modeling approach that first tokenizes sensor observations with VQVAE, then predicts the future via discrete diffusion. To efficiently decode and denoise tokens in parallel, we recast Masked Generative Image Transformer as discrete diffusion and enhance it with a few simple changes, resulting in notable improvement. When applied to learning world models on point cloud observations, Copilot4D reduces prior SOTA Chamfer distance by more than **65**% for 1s prediction, and more than **50**% for 3s prediction, across NuScenes, KITTI Odometry, and Argoverse2 datasets. Our results demonstrate that discrete diffusion on tokenized agent experience can unlock the power of GPT-like unsupervised learning for robotics.

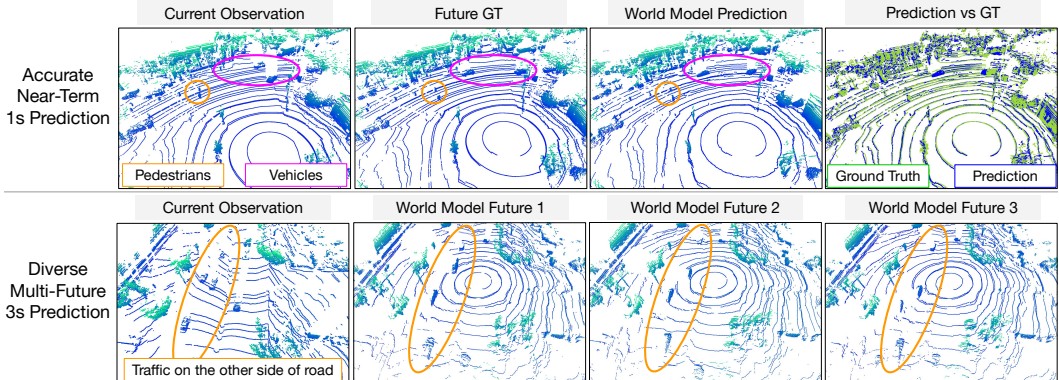

Figure 1: **Our unsupervised world model Copilot4D** can produce accurate near-term 1s predictions and diverse multi-future 3s predictions directly on the level of point cloud observations.

## 1 INTRODUCTION

World models explicitly represent the knowledge of an autonomous agent about its environment. They are defined as a generative model that predicts the next observation in an environment given past observations and the current action. Such a generative model can learn from any unlabeled agent experience, and can be used for both learning and planning in the model-based reinforcement learning framework (Sutton, 1991). This approach has excelled in domains such as Atari (Kaiser et al., 2019), robotic manipulation (Nagabandi et al., 2020), and Minecraft (Hafner et al., 2023).

Learning world models can be viewed as a special case of sequence modeling on agent experience. While Generative Pre-trained Transformers (GPT) (Brown et al., 2020) have enabled rapid progress

in natural language processing (NLP) via sequence modeling of unlabeled text corpus, progress for scaling world models has been less rapid in robotic applications such as autonomous driving. Prediction systems in autonomous driving still require supervised learning, either on the level of bounding boxes (Luo et al., 2018), semantic segmentation (Sadat et al., 2020), or instance segmentation (Hu et al., 2021). However, just as GPT learns to understand language via next token prediction, if a world model can predict unlabeled future observations really well, it must have developed a general understanding of the scene including geometry and dynamics. We ask: *what makes it difficult to learn an unsupervised world model that directly predicts future observations*?

**(i)** The observation space can be complex and unstructured. Whether it is autonomous driving or robotic hands solving Rubik's cubes (Akkaya et al., 2019), selecting a loss on the observation space and building a generative model that captures meaningful likelihoods can be highly non-trivial. By contrast, in natural language processing, language models like GPT (Brown et al., 2020) first tokenize a text corpus, then predict discrete indices like a classifier, leading to impressive success. Fortunately, this gap can addressed by training a VQVAE-like (Van Den Oord et al., 2017) model to tokenize any inputs, from images (Ramesh et al., 2021) to point clouds (Xiong et al., 2023).

**(ii)** The generative model needs to be scalable. In particular, language models are known to scale well (Kaplan et al., 2020), but they only decode one token at a time. In domains such as autonomous driving, a single observation has tens of thousands of tokens, so parallel decoding of tokens becomes a must. On the other hand, decoding all the tokens of an observation in parallel, which is sufficient for achieving success in Minecraft (Hafner et al., 2023), would incorrectly assume that all those tokens are conditionally independent given past observations. Thanks to Masked Generative Image Transformer (MaskGIT) (Chang et al., 2022), we can train a model to iteratively decode an arbitrary number of tokens in parallel. In this work, we recast MaskGIT into the discrete diffusion framework (Austin et al., 2021), resulting in a few simple changes that notably improve upon MaskGIT.

The analysis above sheds light on a scalable approach to building world models: tokenize each observation frame with VQVAE, apply discrete diffusion on each frame, and autoregressively predict the future. We apply this approach to the task of point cloud forecasting in autonomous driving (Weng et al., 2021; Mersch et al., 2022; Khurana et al., 2023), which aims to predict future point cloud observations given past observations and future ego vehicle poses. This task is essentially about building an unsupervised world model on Lidar sensor observations. We design our neural architectures to leverage prior knowledge for this task: our tokenizer uses an implicit representation for volume rendering (Mildenhall et al., 2021) of ray depth in the VQVAE decoder; the world model uses a Transformer that interleaves spatial (Liu et al., 2021) and temporal blocks in Bird-Eye View (BEV). After tokenization, our world model operates entirely on discrete token indices.

Our approach, named Copilot4D, significantly outperforms prior state-of-the-art for point cloud forecasting in autonomous driving. On NuScenes (Caesar et al., 2020), KITTI Odometry (Geiger et al., 2012), and Argoverse2 (Wilson et al., 2023) datasets, **Copilot4D reduces prior SOTA Chamfer distance by $65\% - 75\%$ for 1s prediction, and more than $50\%$ for 3s prediction.** In Figure 1, we showcase that not only is our world model able to make accurate predictions on a 1s time horizon, it has also managed to learn the multi-modality of future observations on a 3s time horizon. The results validate our analysis that the combination of tokenization and discrete diffusion can unlock the possibility of learning world models at scale on real-world data.

## 2 RELATED WORK

**World Models** predict the next observation in an environment given the current action and the past observations. The idea of learning a world model from data dates back to adaptive control (Slotine et al., 1991), which applies parameter estimation to a fixed structure of the dynamics. Under model-based reinforcement learning frameworks such as Dyna (Sutton, 1991), many attempts have been made to use deep generative models as world models. Ha & Schmidhuber (2018) trained a VAE (Kingma & Welling, 2013) to encode observations, and a recurrent neural net (RNN) on the latent codes to model the dynamics. Dreamer-v2 (Hafner et al., 2020) finds that replacing Gaussian latents with discrete latents significantly improves world modeling for Atari. IRIS (Micheli et al., 2022) shows that using a Transformer (Vaswani et al., 2017) rather than an RNN for dynamics modeling further improves Atari results. Those prior works provide a valuable guide for building world models for autonomous driving; we tackle the point-cloud forecasting task (Weng et al., 2021; Mersch et al., 2022; Weng et al., 2022; Khurana et al., 2023) using lessons learned from those other domains.

**Diffusion Models** are a class of generative models that define a forward process from data distribution to noise distribution in closed-form, and then learn the reverse process from noise distribution to data distribution (Sohl-Dickstein et al., 2015). Diffusion for continuous random variables typically uses Gaussian noise, and utilizes properties of Gaussian distributions to simplify the training objectives (Ho et al., 2020; Kingma et al., 2021) and speed up inference (Song et al., 2020). In comparison, diffusion for discrete data (Austin et al., 2021) has received less attention from the community. Recently, Masked Generative Image Transformer (MaskGIT) (Chang et al., 2022) shows that training a BERT (Devlin et al., 2018) on image tokens with an aggressive masking schedule can outperform Gaussian diffusion. MaskGIT has been successfully applied to many applications such as text-to-image generation (Chang et al., 2023), video prediction (Gupta et al., 2022; Yu et al., 2023), and point cloud generation (Xiong et al., 2023). Our work sheds light on the connection between MaskGIT and discrete diffusion, and how MaskGIT can be further improved based on diffusion.

**3D Representation for Point Clouds** has long been studied in both robot perception and 3D scene generation. For self-driving perception, point cloud data from Lidar sensor typically plays an important role. Modern approaches such as VoxelNet (Zhou & Tuzel, 2018) and PointPillars (Lang et al., 2019) first apply a PointNet (Qi et al., 2017) on each voxel or pillar, typically followed by 2D convolution in Bird-Eye View (BEV) (Yang et al., 2018) for tasks such as 3D object detection. For 3D generation, implicit neural scene representation has been gaining popularity since Neural Radiance Fields (NeRF) (Mildenhall et al., 2021); with an implicit function for occupancy, point clouds can be obtained through differentiable depth rendering (Rematas et al., 2022; Yang et al., 2023). This representation is also used in prior work on point cloud forecasting (Khurana et al., 2023). Our tokenizer draws ideas from 3D detection to design the encoder, and from implicit neural scene representation to design the decoder. After tokenization, however, the specific 3D representations are abstracted away from the world model, which operates on discrete tokens.

## 3 BACKGROUND: DIFFUSION MODELS

We review diffusion for a single random variable $x_0$ (which can be trivially extended to multi-variate $\mathbf{x}_0$). Given $x_0$ and the forward process $q(x_{1:K}|x_0)$, diffusion typically learns a reverse process $p_\theta(x_{k-1}|x_k)$ by maximizing an evidence-lower bound (ELBO) $\log p_\theta(x_0) \geq -\mathcal{L}_{elbo}(x_0, \theta) =$

$$\mathbb{E}_{q(x_{1:K}|x_0)}\Big[-\sum_{k>1} D_{KL}(q(x_{k-1}|x_k, x_0) \parallel p_\theta(x_{k-1}|x_k)) + \log p_\theta(x_0|x_1) - D_{KL}(q(x_K|x_0) \parallel p(x_K))\Big]$$

For discrete random variables, since we can easily sum over its probabilities, $p_\theta(x_{k-1}|x_k)$ is often parameterized to directly infer $x_0$, as done in D3PM (Austin et al., 2021):

$$p_\theta(x_{k-1} \mid x_k) = \sum_{x_0} q(x_{k-1} \mid x_k, x_0) p_\theta(x_0 \mid x_k) \tag{1}$$

Calculating the posterior $q(x_{k-1}|x_k, x_0) = q(x_{k-1}|x_0)q(x_k|x_{k-1})/q(x_k|x_0)$ is necessary in the original diffusion loss, which means that the cumulative forward transition matrix defined in $q(x_k|x_0)$ often requires a closed-form solution. In D3PM, absorbing diffusion recasts BERT (Devlin et al., 2018) as a one-step diffusion model, where the forward diffusion process gradually masks out ground-truth tokens. VQ-Diffusion (Gu et al., 2022) points out that, when $x_k \neq x_0$, the posterior in absorbing diffusion is not well-defined since in that case $q(x_k|x_0) = 0$, which motivated adding uniform diffusion in their model. By contrast, MaskGIT (Chang et al., 2022) has significantly simpler training and sampling procedures based on BERT alone: for training, it masks a part of the input tokens (with an aggressive masking schedule) and then predicts the masked tokens from the rest; for sampling, it iteratively decodes tokens in parallel based on predicted confidence.

Classifier-free diffusion guidance (Ho & Salimans, 2022) has become a standard tool for diffusion-based conditional generation. Given context $c$, it has been shown that sampling from $\tilde{p}_\theta(x_0|x_k, c) \propto p_\theta(x_0|x_k, c)(p_\theta(x_0|x_k, c)/p_\theta(x_0|x_k))^w$ rather than directly from $p_\theta(x_0|x_k, c)$ performs significantly better. Chang et al. (2023) has proposed directly modifying the logits of MaskGIT:

$$\text{logits}_{\text{cfg}}(\tilde{x}_0|x_{k+1}, c) = \text{logits}(\tilde{x}_0|x_{k+1}, c) + w \cdot (\text{logits}(\tilde{x}_0|x_{k+1}, c) - \text{logits}(\tilde{x}_0|x_{k+1})) \tag{2}$$

in order to perform classifier-free guidance analogous to its counterpart in diffusion.

## 4 METHOD: COPILOT4D

Given a sequence of agent experience $(\mathbf{o}^{(1)}, \mathbf{a}^{(1)}, \cdots, \mathbf{o}^{(T-1)}, \mathbf{a}^{(T-1)}, \mathbf{o}^{(T)})$ where $\mathbf{o}$ is an observation and $\mathbf{a}$ is an action, we aim to learn a world model $p_\theta$ that predicts the next observation given

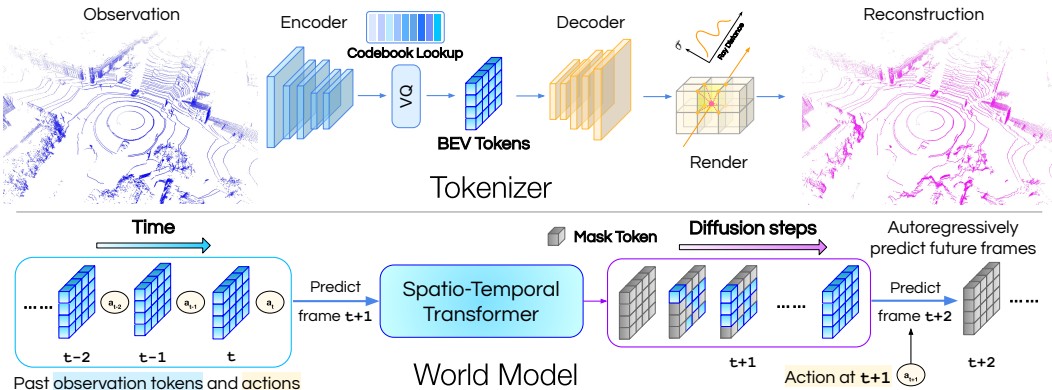

Figure 2: **An overview of our method for Copilot4D**, which first tokenizes sensor observations with a VQVAE-like tokenizer, then predicts the future via discrete diffusion. The tokenizer encodes point clouds into discrete latents in Bird-Eye View (BEV), and does reconstruction via differentiable depth rendering. The world model is a discrete diffusion model that operates on BEV tokens.

the past observations and actions. In the autonomous driving setting that we tackle, the observations $\{\mathbf{o}^{(t)}\}_t$ are point clouds from the Lidar sensor, and the actions $\{\mathbf{a}^{(t)}\}_t$ are $SE(3)$ poses of the ego vehicle. We first tokenize each observation $\mathbf{o}^{(t)}$ into $\mathbf{x}^{(t)} \in \{0, \cdots, |V|-1\}^N$, where $N$ is the number of tokens in each observation, and $V$ is the vocabulary defined by the learned codebook in VQVAE. We denote $\mathbf{x}^{(t)}$ as the tokenized observation $t$. The learning objective is:

$$\arg\max_\theta \sum_t \log p_\theta(\mathbf{x}^{(t)} \mid \mathbf{x}^{(1)}, \mathbf{a}^{(1)}, \cdots, \mathbf{x}^{(t-1)}, \mathbf{a}^{(t-1)}) \tag{3}$$

Our world model is a discrete diffusion model (Austin et al., 2021; Lezama et al., 2023) that is able to perform conditional generation given past observations and actions. We denote $\mathbf{x}_k^{(t)}$ as the tokenized observation $t$ under forward diffusion step $k$. $k = 0$ is the original data distribution, and the total number of steps $K$ can be arbitrary at inference. We outline the inference process as follows: to predict an observation at timestep $t+1$, the world model first tokenizes past observations $\mathbf{o}^{(1)} \cdots \mathbf{o}^{(t)}$ into $\mathbf{x}^{(1)} \cdots \mathbf{x}^{(t)}$, applies discrete diffusion for next frame prediction to decode the initially fully masked $\mathbf{x}_K^{(t+1)}$ into fully decoded $\mathbf{x}_0^{(t+1)}$, and then passes $\mathbf{x}_0^{(t+1)}$ into the decoder of the tokenizer to render the next observation $\mathbf{o}^{(t+1)}$. For a visual overview of our method, see Figure 2.

## 4.1 TOKENIZE THE 3D WORLD

We propose a novel VQVAE-like (Van Den Oord et al., 2017) model to tokenize the 3D world represented by point clouds (Xiong et al., 2023). The model learns latent codes in Bird-Eye View (BEV) and is trained to reconstruct point clouds via differentiable depth rendering.

The encoder uses standard components from point-cloud based object detection literature: first, aggregate point-wise features of each voxel with a PointNet (Qi et al., 2017); second, aggregate voxel-wise features into BEV pillars (Lang et al., 2019); finally, apply a Swin Transformer backbone (Liu et al., 2021) to obtain a feature map that is 8x downsampled from the initial voxel size in BEV. The output of the encoder, $\mathbf{z} = E(\mathbf{o})$, goes through a vector quantization layer to produce $\hat{\mathbf{z}}$.

The novelty of our tokenizer lies in the decoder, which produce two branches of outputs after a few Swin Transformer blocks. The first branch uses an implicit representation (Mildenhall et al., 2021) so that we can query occupancy values at continuous coordinates. To query $(x, y, z)$, we apply bilinear interpolation on a 3D neural feature grid (NFG) (Yang et al., 2023) outputted by the decoder to obtain a feature descriptor, which then goes through a multi-layer perceptron (MLP) and sigmoid to arrive at an occupancy value $\alpha$ in $[0, 1]$. Given a ray $\mathbf{r}(h) = \mathbf{p} + h\mathbf{d}$ starting at point $\mathbf{p}$ and traveling in direction $\mathbf{d}$, the expected depth $D$ can be calculated via differentiable depth rendering on $N_\mathbf{r}$ sampled points $\{(x_i, y_i, z_i)\}_{i=1}^{N_\mathbf{r}}$ along the ray:

$$\alpha_i = \sigma(\text{MLP}(\text{interp}(\text{NFG}(\hat{\mathbf{z}}), (x_i, y_i, z_i)))) \quad w_i = \alpha_i \prod_{j=1}^{i-1}(1-\alpha_j) \quad D(\mathbf{r}, \hat{\mathbf{z}}) = \sum_{i=1}^{N_\mathbf{r}} w_i h_i \tag{4}$$

| **Algorithm 1** Training | **Algorithm 2** Sampling |
|---|---|
| 1: **repeat** | 1: $\mathbf{x}_K$ = all mask tokens |
| 2: $\quad \mathbf{x}_0 : \{1, \cdots, |V|\}^N \sim q(\mathbf{x}_0)$ | 2: **for** $k = K-1, \ldots, 0$ **do** |
| 3: $\quad u_0 \sim \text{Uniform}(0, 1)$ | 3: $\quad \tilde{\mathbf{x}}_0 \sim p_\theta(\cdot \mid \mathbf{x}_{k+1})$ |
| 4: $\quad$ Randomly mask $\lceil \gamma(u_0)N \rceil$ tokens in $\mathbf{x}_0$ | 4: $\quad \boldsymbol{l}_k = \log p_\theta(\tilde{\mathbf{x}}_0 \mid \mathbf{x}_{k+1}) + Gumbel(0,1) \cdot k/K$ |
| 5: $\quad u_1 \sim \text{Uniform}(0, 1)$ | 5: $\quad$ On non-mask indices of $\mathbf{x}_{k+1}$: $\boldsymbol{l}_k \leftarrow +\infty$ |
| 6: $\quad$ Randomly noise $(u_1 \cdot \eta)\%$ of remaining tokens | 6: $\quad M = \lceil \gamma(k/K)N \rceil$ |
| 7: $\quad \mathbf{x}_k \leftarrow$ *masked-and-noised* $\mathbf{x}_0$ | 7: $\quad \mathbf{x}_k \leftarrow \tilde{\mathbf{x}}_0$ on top-$M$ indices of $\boldsymbol{l}_k$ |
| 8: $\quad \arg\max_\theta \log p_\theta(\mathbf{x}_0 \mid \mathbf{x}_k)$ with cross entropy | 8: **end for** |
| 9: **until** converged | 9: **return** $\mathbf{x}_0$ |

Figure 3: Our improved discrete diffusion algorithm. Differences with MaskGIT (Chang et al., 2022) are highlighted in blue. $\gamma(u) = \cos(u\pi/2)$ is the mask schedule. We set $\eta = 20$ by default.

The second branch learns a coarse reconstruction of the point clouds by predicting whether a voxel has points in its inputs. We denote this binary probability as $\mathbf{v}$. During inference, this branch is used for spatial skipping (Li et al., 2023) to speed up point sampling in rendering.

The loss function for the tokenizer is a combination of the vector quantization loss $\mathcal{L}_{\text{vq}}$ and the rendering loss $\mathcal{L}_{\text{render}}$. The vector quantization loss learns the codebook and regularizes the latents: $\mathcal{L}_{\text{vq}} = \lambda_1 \|\text{sg}[E(\mathbf{o})] - \hat{\mathbf{z}}\|_2^2 + \lambda_2 \|\text{sg}[\hat{\mathbf{z}}] - E(\mathbf{o})\|_2^2$. In the rendering loss, supervision is applied on both branches: the depth rendering branch has an $L1$ loss on depth with an additional term that encourages $w_i$ to concentrate within $\epsilon$ of the surface (Yang et al., 2023); the spatial skipping branch optimizes binary cross entropy. The tokenizer is trained end-to-end to reconstruct the observation:

$$\mathcal{L}_{\text{render}} = \mathbb{E}_\mathbf{r}\left[ \|D(\mathbf{r}, \hat{\mathbf{z}}) - D_{gt}\|_1 + \sum_i \mathbb{1}(|h_i - D_{gt}| > \epsilon)\|w_i\|_2 \right] + \text{BCE}(\mathbf{v}, \mathbf{v}_{gt}) \tag{5}$$

With a pretrained tokenizer, both the inputs and the outputs of the world model are discrete tokens.

## 4.2 MaskGIT as a Discrete Diffusion Model

Masked Generative Image Transformer (MaskGIT) (Chang et al., 2022) has been shown to scale for a variety of applications. Interestingly, discrete diffusion models such as D3PM (Austin et al., 2021) have not yet seen similar success, despite having a much more first-principled framework and toolbox. We observe that the key to recasting MaskGIT as a discrete diffusion model is the following proposition in Campbell et al. (2022). It turns out that the parameterization introduced in Equation (1) allows a further lower bound on ELBO under data distribution $q(x_0)$ (also see A.4),

$$\mathbb{E}_{q(x_0)}[\log p_\theta(x_0)] \geq \mathbb{E}_{q(x_0)}[-\mathcal{L}_{elbo}(x_0, \theta)] \geq \sum_{k=1}^{K} \mathbb{E}_{q(x_0)q(x_k|x_0)}[\log p_\theta(x_0 \mid x_k)] + C \tag{6}$$

Which is almost the same loss as MaskGIT loss, except that: for the diffusion posterior $q(x_k|x_0)$ to be well-defined when $x_k \neq x_0$, uniform diffusion in non-masked locations is needed; and the loss is applied not just to masked locations. This implies that a few simple changes can turn MaskGIT into an *absorbing-uniform* discrete diffusion model. During training, after masking a random proportion of tokens in $x_0$, we inject up to $\eta\%$ of uniform noise into the remaining tokens, and apply a cross entropy loss to reconstruct $x_0$. $\eta$ is a fixed hyper-parameter. During sampling, besides parallel decoding, we allow the model to iteratively denoise earlier sampled tokens. The differences with MaskGIT are highlighted in Algorithms 1 and 2. For conditional generation, classifier-free diffusion guidance can be applied by modifying the logits of $p_\theta$ for sampling $\tilde{x}_0$ and calculating $\boldsymbol{l}_k$ according to Equation (2). While resampling tokens has been known to help MaskGIT (Lezama et al., 2022; 2023), our method only requires training a single model rather than two separate ones.

We now use our discrete diffusion algorithm to build a world model on top of observation tokens.

## 4.3 Learning a World Model

For an autonomous agent, the environment can be viewed as a black box that receives an action and outputs the next observation. A world model is a learned generative model that can be

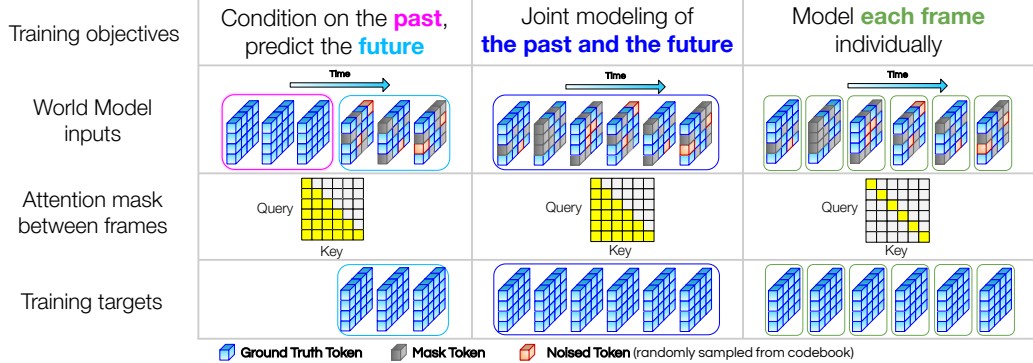

Figure 4: **The training objectives of our world model in Copilot4D**. Other than future prediction, we also train the model to jointly model the past and the future, and individually model each frame. Joint modeling ensures that the model can accurately predict the future even with imperfect past conditioning. Individual frame modeling is necessary for classifier-free diffusion guidance. During training, which objective to optimize is randomly sampled at each iteration.

used in place of the environment. Let $\boldsymbol{\tau} = (\mathbf{x}^{(1)}, \mathbf{a}^{(1)}, \cdots, \mathbf{x}^{(T)})$. Given the context $\mathbf{c}^{(t-1)} = (\mathbf{x}_0^{(1)}, \mathbf{a}^{(1)}, \cdots, \mathbf{x}_0^{(t-1)}, \mathbf{a}^{(t-1)})$ as past agent history, a discrete diffusion world model $p_\theta$ learns to predict the next observation $\mathbf{x}_0^{(t)}$, starting from fully masked $\mathbf{x}_K^{(t)}$, going through $K$ intermediate steps $\mathbf{x}_K^{(t)} \to \mathbf{x}_{K-1}^{(t)} \cdots \to \mathbf{x}_0^{(t)}$ during the reverse process of diffusion. Using Equation (6),

$$\mathbb{E}_{q(\boldsymbol{\tau})}\Big[\underbrace{\sum_{t=1}\log p_\theta(\mathbf{x}_0^{(t)} \mid \mathbf{c}^{(t-1)})}_{\text{Autoregressive future prediction}}\Big] \geq \mathbb{E}_{q(\boldsymbol{\tau})}\Big[\underbrace{\sum_{t=1}\sum_{k=1}\mathbb{E}_{q(\mathbf{x}_k^{(t)}|\mathbf{x}_0^{(t)})}[\log p_\theta(\mathbf{x}_0^{(t)} \mid \mathbf{x}_k^{(t)}, \mathbf{c}^{(t-1)})]}_{\text{Discrete diffusion on each observation}} + C\Big]$$

$$(7)$$

However, in the GPT-like formulation of autoregressive modeling, the model is always able to see all past ground-truth tokens for next frame prediction during training. In robotics, depending on the discretization of time, the world might only change incrementally within the immediate next frame; learning to predict only the immediate next observation will not necessarily lead to long-horizon reasoning abilities even if the loss is optimized well. Therefore, training the world model should go beyond next observation prediction and instead predict an entire segment of future observations given the past. Accordingly, we design the world model to be similar to a spatio-temporal version of BERT, with causal masking in the temporal dimension. Future prediction is done via masking, infilling, and further denoising. The model is trained with a mixture of objectives (see Figure 4):

1. 50% of the time, **condition on the past, denoise the future.**
2. 40% of the time, **denoise the past and the future jointly.**
3. 10% of the time, **denoise each frame individually, regardless of past or future.**

The first objective is about future prediction. The second objective also has a future prediction component, but jointly models the future and the past, resulting in a harder pretraining task. The third objective aims to learn an unconditional generative model, which is necessary for applying classifier-free diffusion guidance during inference. By the word *denoise*, we are referring to Algorithm 1, where parts of the inptus are first masked and noised, and the model learns to reconstruct the original inputs with a cross-entropy loss. All three objectives can be viewed as maximizing the following:

$$\mathbb{E}_{\substack{q(\boldsymbol{\tau}), k_1, \cdots, k_T \sim \text{SampleObj}(\cdot) \\ q(\mathbf{x}_{k_1}^{(1)}|\mathbf{x}_0^{(1)}), \cdots q(\mathbf{x}_{k_T}^{(T)}|\mathbf{x}_0^{(T)})}}\big[\log p_\theta(\ \underbrace{\mathbf{x}_0^{(1)}, \cdots \mathbf{x}_0^{(t-1)}}_{\text{Ignored for Objective type 1}},\ \mathbf{x}_0^{(t)}, \cdots \mathbf{x}_0^{(T)} \mid \mathbf{x}_{k_1}^{(1)}, \cdots \mathbf{x}_{k_T}^{(T)}, \mathbf{a}^{(1)}, \cdots \mathbf{a}^{(T-1)})\big]$$

During inference, we still autoregressively predict one frame at a time. Each frame is sampled using Algorithm 2 with classifier-free diffusion guidance (CFG) in Equation (2). At each timestep $t$, the context in diffusion guidance is $\mathbf{c}^{(t-1)}$, the past observation and action history of the agent. See Figure 10 in the Appendix for an illustration of how CFG is used in our world model.

Next, we outline how both training (with our mixture of objectives) and inference (with classifier-free diffusion guidance) can be implemented with a spatio-temporal Transformer.

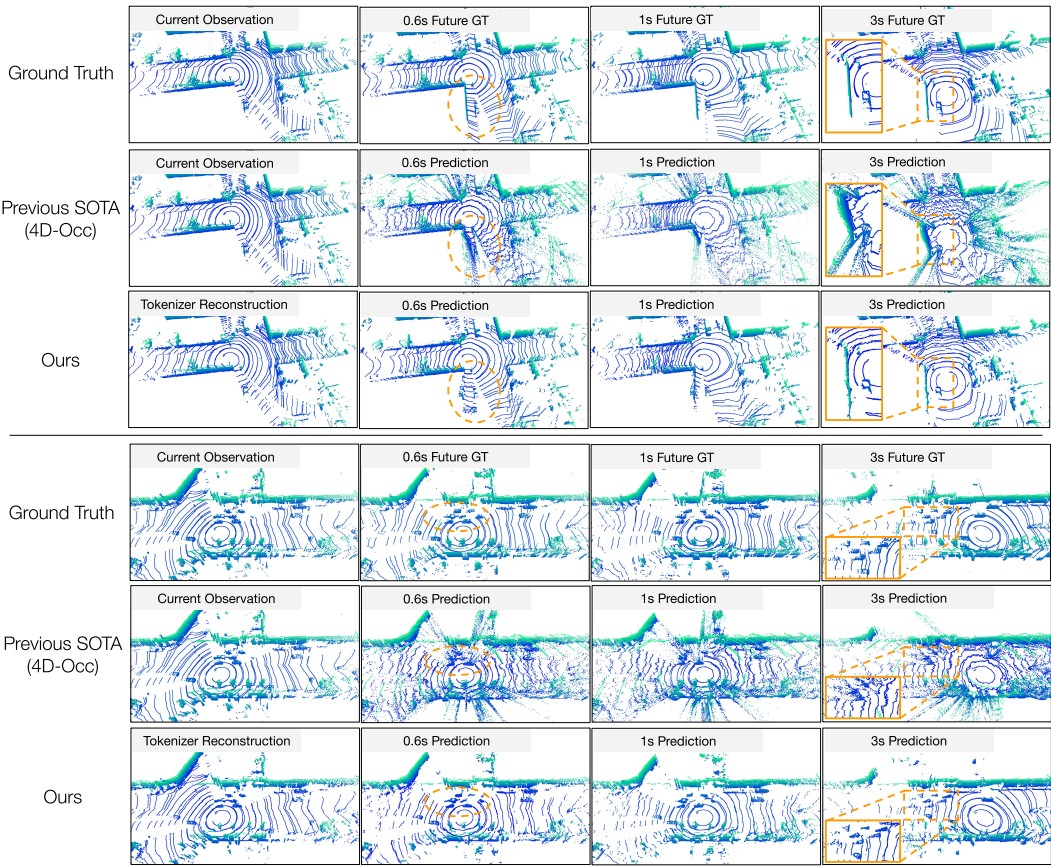

Figure 5: **Qualitative comparison against prior state-of-the-art** method 4D Occupancy (4D-Occ) on Argoverse2 Lidar dataset. Copilot4D achieves significantly better results, demonstrating greater capabilities on novel-view synthesis of an environment as the ego vehicle moves, understanding the motion of other vehicles in the scene, and modeling the Lidar pattern of ground points.

### 4.4 A Spatio-Temporal Transformer for World Modeling

The architecture of our world model is a spatio-temporal Transformer that simply interleaves spatial attention and temporal attention. For spatial attention, we use Swin Transformer (Liu et al., 2021) on each individual frame. For temporal attention, we use GPT2 blocks (Radford et al., 2019) to attend over the same feature location across time. We use a U-Net (Ronneberger et al., 2015) structure that combine three levels of feature with residual connections, and make predictions at the same resolution as the initial inputs. Actions, which in our case are the poses of the ego vehicle, are added to the beginning of each feature level corresponding to their observations, after being flattened and going through two linear layers with LayerNorm (Ba et al., 2016) in between.

Temporal attention mask plays a crucial role in both training and inference. During training, when optimizing the first two types of objective, causal masking is applied to all temporal Transformer blocks; when optimizing the third type of objective to learn an unconditional generative model, the temporal attention mask becomes an identity matrix such that each frame can only attend to itself. During inference, the model decodes and denoises one frame at a time; classifier-free diffusion guidance can be efficiently implemented by increasing temporal sequence length by 1, and setting the attention mask to be a causal mask within the previous sequence length, and an identity mask for the last frame, so that this added frame becomes unconditional generation (see Figure 10).

## 5 Experiments

In this section, we aim to answer the following questions: **(1)** Can our proposed world modeling approach outperform previous state-of-the-art point cloud forecasting methods on large-scale self-

Table 1: **Results on NuScenes and KITTI Odometry datasets.** Comparison against previous state-of-the-art methods: SPFNet (Weng et al., 2021), S2Net (Weng et al., 2022), ST3DCNN (Mersch et al., 2022), and 4D Occupancy (4D-Occ) (Khurana et al., 2023). The color magenta means that a metric is computed within the Region of Interest defined in Khurana et al. (2023): $-70$m to $+70$m in both $x$-axis and $y$-axis, $-4.5$m to $+4.5$m in $z$-axis. L1 Med means median L1 error within ROI; AbsRel Med means the median absolute relative L1 error percentage within ROI.

| NuScenes 1s | Chamfer↓ | L1 Med↓ | AbsRel Med↓ | L1 Mean↓ | AbsRel↓ | Chamfer↓ |
|---|---|---|---|---|---|---|
| SPFNet | 2.24 | - | - | 4.58 | 34.87 | 4.17 |
| S2Net | 1.70 | - | - | 3.49 | 28.38 | 2.75 |
| 4D-Occ | 1.41 | 0.26 | 4.02 | 1.40 | 10.37 | 2.81 |
| Copilot4D | **0.36** | **0.10** | **1.30** | **1.30** | **8.58** | **2.01** |
| **NuScenes 3s** | | | | | | |
| SPFNet | 2.50 | - | - | 5.11 | 32.74 | 4.14 |
| S2Net | 2.06 | - | - | 4.78 | 30.15 | 3.47 |
| 4D-Occ | 1.40 | 0.43 | 6.88 | 1.71 | 13.48 | 4.31 |
| Copilot4D | **0.58** | **0.14** | **1.86** | **1.51** | **10.38** | **2.47** |
| **KITTI 1s** | | | | | | |
| ST3DCNN | 4.11 | - | - | 3.13 | 26.94 | 4.51 |
| 4D-Occ | 0.51 | 0.20 | 2.52 | 1.12 | 9.09 | 0.61 |
| Copilot4D | **0.18** | **0.11** | **1.32** | **0.95** | **8.59** | **0.21** |
| **KITTI 3s** | | | | | | |
| ST3DCNN | 4.19 | - | - | 3.25 | 28.58 | 4.83 |
| 4D-Occ | 0.96 | 0.32 | 3.99 | 1.45 | 12.23 | 1.50 |
| Copilot4D | **0.45** | **0.17** | **2.18** | **1.27** | **11.50** | **0.67** |

Table 2: **Results on Argoverse 2 Lidar Dataset.** We evaluate on evenly subsampled 4000 frames on the Argoverse 2 Lidar validation set. All metrics are computed within the ROI.

| 1s Prediction | Chamfer↓ | L1 Med↓ | AbsRel Med↓ | L1 Mean↓ | AbsRel Mean↓ |
|---|---|---|---|---|---|
| 4D-Occ | 1.42 | 0.24 | 1.67 | 2.04 | 11.02 |
| Copilot4D | **0.26** | **0.15** | **0.94** | **1.61** | **8.75** |
| **3s Prediction** | | | | | |
| 4D-Occ | 1.99 | 0.42 | 2.88 | 2.62 | 15.66 |
| Copilot4D | **0.55** | **0.19** | **1.26** | **1.99** | **11.86** |

driving datasets? **(2)** How important is classifier-free diffusion guidance in a discrete diffusion world model? **(3)** Does our improved discrete diffusion algorithm achieve better performance compared to MaskGIT, in terms of learning a world model?

**Datasets and Experiment Setting**: We use NuScenes (Caesar et al., 2020), KITTI Odometry (Geiger et al., 2012), and Argoverse2 Lidar (Wilson et al., 2023), three commonly used large-scale datasets for autonomous driving. Our evaluation protocol follows Khurana et al. (2023): on each dataset, we evaluate 1s prediction and 3s prediction by training two models. Each model is given past point cloud observations and future poses (which are the *actions*) of the ego vehicle. For KITTI and Argoverse2, each model receives 5 past frames and outputs 5 future frames (spanning either 1s or 3s). For NuScenes, the 2Hz dataset is used; as a result, 1s prediction takes in two past frames and outputs two future frames; 3s prediction takes in 6 past frames and outputs 6 future frames. While our world modeling approach is not limited to this experimental setting, we follow the same protocol to be able to directly compare against prior methods.

**Metrics**: we follow the common metrics for point cloud forecasting (Khurana et al., 2023), which include Chamfer distance, L1 depth for raycasting (L1 Mean), and relative L1 error ratio (AbsRel). However, we notice an issue with the previously proposed metrics: while model predictions are made only within the region of interest (ROI), the ground-truth point clouds are not cropped accord-

Table 3: Our method for classifier-free diffusion guidance (CFG) significantly improves prediction results and especially the Chamfer Distance metric.

| NuScenes 3s | Chamfer↓ | L1 Med↓ | AbsRel Med↓ | L1 Mean↓ | AbsRel Mean↓ |
|---|---|---|---|---|---|
| $w = 0.0$ *(no CFG)* | 1.40 | **0.13** | 1.81 | 1.23 | **8.34** |
| $w = 1.0$ | **0.56** (60% ↓) | **0.13** | **1.78** | **1.22** | 9.32 |
| $w = 2.0$ | 0.58 | 0.14 | 1.86 | 1.27 | 9.90 |

Table 4: Our proposed discrete diffusion algorithm significantly improves upon previous masked modeling method MaskGIT (Chang et al., 2022). Both models use only 10 sampling steps per frame of prediction ($128 \times 128$ tokens), applied with classifier-free diffusion guidance $w = 2.0$.

| NuScenes 3s | Chamfer↓ | L1 Med↓ | AbsRel Med↓ | L1 Mean↓ | AbsRel Mean↓ |
|---|---|---|---|---|---|
| MaskGIT | 0.82 | 0.16 | 2.09 | 1.41 | 11.66 |
| Ours | **0.58** (29% ↓) | **0.14** | **1.86** | **1.27** | **9.90** |

ing to the ROI, resulting in artifically high error metrics simply because the ROI might not cover the full point cloud range. Consequently, we report metrics computed within the ROI in magenta, which better reflects the performance of a model. We also report the median of L1 depth error (L1 Med), since the median is more robust to outliers than the mean. Following previous evaluation protocols, the ROI is defined to be $-70$m to $+70$m in both $x$-axis and $y$-axis, $-4.5$m to $+4.5$m in $z$-axis around the ego vehicle.

**Benchmark against state-of-the-art**: Table 1 and 2 show quantitative comparisons with state-of-the-art unsupervised point cloud forecasting methods on three datasets. Our baselines include SPFNet (Weng et al., 2021), S2Net (Weng et al., 2022), ST3DCNN (Mersch et al., 2022), and 4D Occupancy (Khurana et al., 2023). Our method Copilot4D is able to outperform prior methods by a significant margin across all three datasets. In particular, for 1s prediction, we are able to see a $65\% - 75\%$ reduction in Chamfer Distance compared to prior SOTA across all three datasets; for 3s prediction, we are able to see more than $50\%$ reduction in Chamfer. We also present a qualitative comparison with previous SOTA (4D Occupancy) in Figure 5. Copilot4D learns qualitatively better future predictions, demonstrating an impressive ability to synthesize novel views on the background as the ego vehicle moves and forecast the motion of other vehicles.

**Classifier-free diffusion guidance (CFG) is important**: in Table 3, we show that CFG reduces Chamfer by as much as 60% in an ablation on NuScenes 3s prediction. The result is not surprising considering that CFG has become a standard tool in text-to-image generation; here we show that using past agent history ($\mathbf{c}^{(t-1)}$ in Section 4.3) as CFG conditioning improves world modeling. Intuitively, CFG amplifies the contribution of the conditioned information in making predictions.

**Improvement upon MaskGIT**: Our proposed simple changes to MaskGIT train the model to do a harder denoising task in Algorithm 1 and thus allow the inference procedure in Algorithm 2 to iteratively revise chosen tokens. Those changes notably improve MaskGIT on an ablation run of NuScenes 3s prediction task, shown in Table 4. In our case, each frame has $128 \times 128$ tokens with 10 diffusion steps; on average each diffusion step decodes 1600 new tokens in parallel. Being able to denoise and resample already decoded tokens reduces Chamfer distance of 3s prediction by $29\%$.

## 6 CONCLUSION

Learning unsupervised world models is a promising paradigm where GPT-like pretraining for robotics can potentially scale. In this work, we first identify two practical bottlenecks of this paradigm: simplifying the complex observation space, and building a scalable generative model for spatio-temporal data. We then propose Copilot4D, which combines observation tokenization, discrete diffusion, and the Transformer architecture as a new approach for building unsupervised world models, achieving state-of-the-art results for the point cloud forecasting task in autonomous driving. One particularly exciting aspect of Copilot4D is that it is broadly applicable to many domains. We hope that future work will combine our world modeling approach with model-based reinforcement learning to improve the decision making capabilities of autonomous agents.

# 7 ACKNOWLEDGEMENT

We thank Chris Zhang, Anqi Joyce Yang, Thomas Gilles, and many others on the Waabi team for helpful discussions and valuable support throughout the project.

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

## A  APPENDIX

### A.1  ADDITIONAL QUANTITATIVE RESULTS

Table 5: Zero-shot transfer performance across datasets: training on Argoverse 2 (AV2) Lidar and evaluating on KITTI Odometry. Compared to 4d Occupancy (4d-Occ) (Khurana et al., 2023), our method Copilot4D achieves significantly better dataset transfer results.

| [AV2 → KITTI] 1s Prediction | Chamfer↓ | L1 Mean↓ | AbsRel↓ | Chamfer↓ |
|---|---|---|---|---|
| 4D-Occ | 2.52 | 1.71 | 14.85 | 3.18 |
| Copilot4D | **0.36** | **1.16** | **11.1** | **0.44** |
| [AV2 → KITTI] 3s Prediction | | | | |
| 4D-Occ | 4.83 | 2.52 | 23.87 | 5.79 |
| Copilot4D | **1.12** | **1.91** | **19.0** | **1.65** |

**Zero-shot transfer performance across datasets**: Table 5 follows prior protocols of training on Argoverse2 and testing on KITTI Odometry, and shows that: Copilot4D achieves **more than** $4\times$ **smaller Chamfer distance** compared to 4D Occupancy, on both 1s and 3s prediction. Our results indicate that tokenization, discrete diffusion, and a spatio-temporal Transformer form a powerful combination that can achieve a greater degree of cross-dataset transfer.

Table 6: Results on the test set of Argoverse 2 Lidar dataset. Besides the results on validation set presented in the main paper, we also evaluate on evenly subsampled 4000 frames on the Argoverse 2 Lidar test set. The results are very similar; we present the test-set results here for completeness. All metrics are computed within the ROI.

| 1s Prediction | Chamfer↓ | L1 Med↓ | AbsRel Med↓ | L1 Mean↓ | AbsRel Mean↓ |
|---|---|---|---|---|---|
| 4D-Occ | 1.51 | 0.25 | 1.66 | 2.07 | 11.21 |
| Copilot4D | **0.25** | **0.15** | **0.96** | **1.64** | **9.02** |
| 3s Prediction | | | | | |
| 4D-Occ | 2.12 | 0.45 | 3.05 | 2.69 | 16.48 |
| Copilot4D | **0.61** | **0.20** | **1.29** | **2.08** | **12.49** |

**Ablation studies on the tokenizer**:

- Table 7 provides an ablation on the effect of spatial skipping (Li et al., 2023), where we train another tokenizer from scratch without the coarse reconstruction branch or the spatial skipping process; this tokenizer is only supervised with the depth rendering loss. The table shows that spatial skipping improves point cloud reconstructions. We illustrate the details of the spatial skipping process in Figure 8.

- We also show that differentiable depth rendering using implicit representation is crucial. In Figure 9, we provide a qualitative comparison between the VQVAE in UltraLiDAR (Xiong et al., 2023) and our proposed tokenizer. The UltraLiDAR model only predicts whether a voxel has points present, and is similar to our coarse reconstruction branch. The results show that UltraLiDAR is unable to reconstruct fine-grained geometry due to the limited resolution of voxel predictions, whereas our tokenizer is able to produce high-fidelity point clouds that recover the details in the input point clouds.

### A.2  MODEL DETAILS

Both the tokenizer and the world model are quite lightweight. The tokenizer is a 13-Million parameter model; the world model is a 39-Million parameter model. To put the parameter counts in context, the tokenizer has fewer parameters than a ResNet-34 (He et al., 2016) (21.8 M parameters), and the current world model has fewer parameter count than a ResNet-101 (44.5M parameters). Achieving our results on such a small model scale is another highlight of our algorithm.

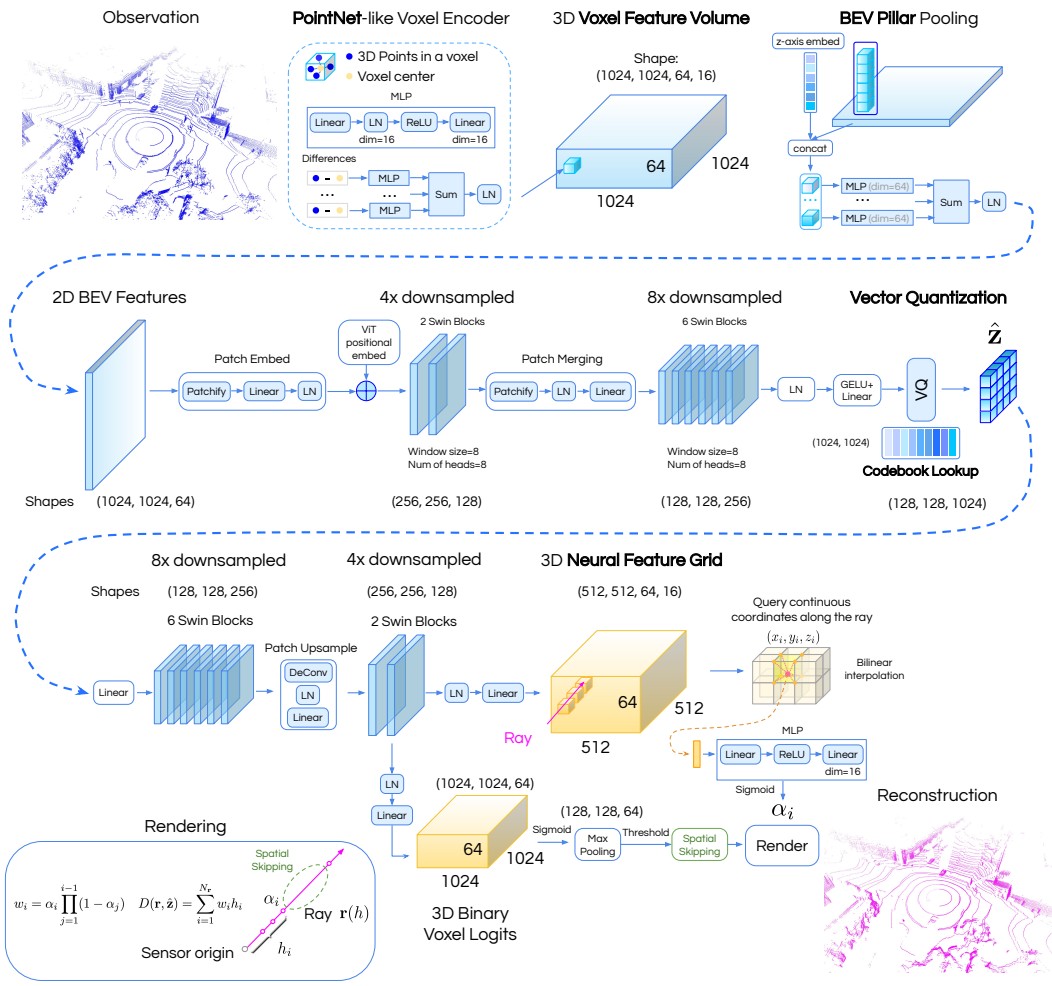

Figure 6: **Detailed architecture of our point cloud tokenizer**, which combines PointNet, Bird-Eye View (BEV) representation, Neural Feature Grid (NFG), and differentiable depth rendering.

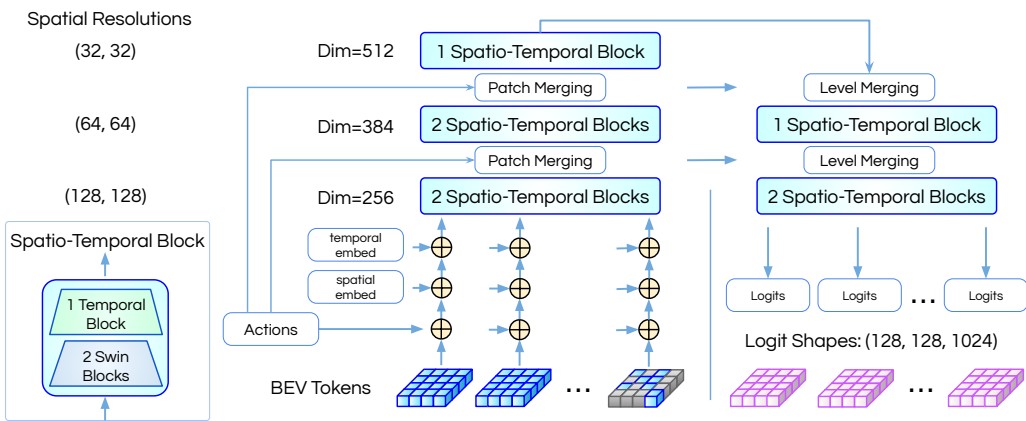

Figure 7: **Detailed architecture of our U-Net based Transformer world model**, which interleaves spatial and temporal Transformer blocks and applies multiple levels of spatial feature resolutions.

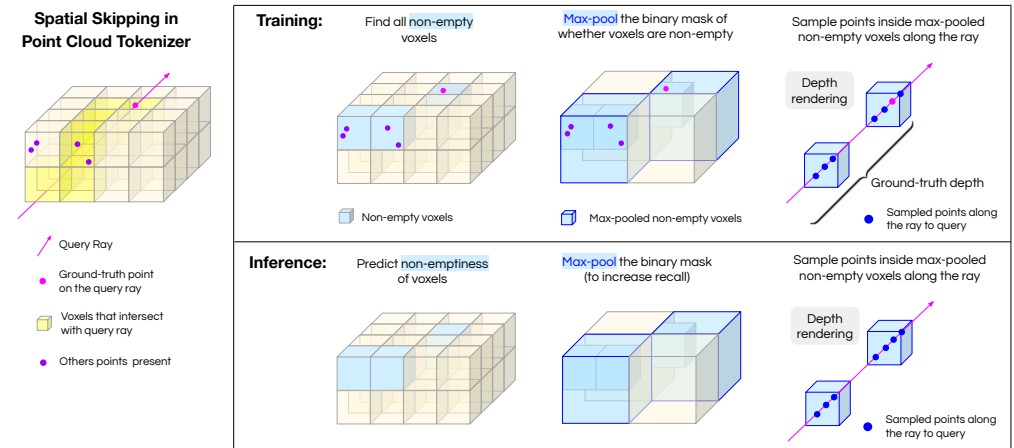

Figure 8: **Illustration of the spatial skipping process.** We find that spatial skipping (Li et al., 2023) not only speeds up tokenizer inference but also results in slightly better results. Rather than uniformly sampling points along the ray for depth rendering, we sample points only within the estimated non-empty regions along the ray. Specifically, we predict a binary mask indicating which voxels are non-empty, apply max pooling on this binary mask to increase recall of non-empty voxels, and sample (ray) points only within the max-pooled non-empty voxels. During training, the ground-truth binary mask of non-emptiness can be computed from original point clouds; this mask is also used to supervise the coarse reconstruction branch. During inference, we threshold the prediction from the coarse reconstruction branch to produce the mask.

| NuScenes [Reconstruction] | Chamfer (within ROI)↓ | L1 Median↓ | L1 Mean↓ | Chamfer↓ |
|---|---|---|---|---|
| No spatial skipping | 0.148 | **0.044** | 1.42 | 1.66 |
| With spatial skipping | **0.082** | **0.044** | **0.82** | **1.64** |

Table 7: **Ablation study on spatial skipping in the tokenizer**. Here we train another tokenizer from scratch without the coarse reconstruction branch or the spatial skipping process. Our results show that spatial skipping leads to improved reconstructions, which is likely due to the fact that it allows the sampling of points along the rays to focus on the non-empty regions.

### A.2.1 TOKENIZER

The tokenizer follows a VQVAE (Van Den Oord et al., 2017) structure to encode point clouds into Bird-Eye View (BEV) tokens and reconstruct input point clouds via differentiable depth rendering.

**Tokenizer encoder** The initial layer in the encoder is a voxel-wise PointNet (Qi et al., 2017) similar to VoxelNet (Zhou & Tuzel, 2018) that encodes the distance of each point to its corresponding voxel center, with one modification to PointNet: while the initial PointNet uses max pooling as the permutation-invariant aggregation function, we use a sum operation + Layer-Norm (Ba et al., 2016), which is also a permutation-invariant function. We use a voxel size of $15.625\text{cm} \times 15.625\text{cm} \times 14.0625\text{cm}$ in the $x, y, z$ dimensions, following an input voxel size similar to the one used in Xiong et al. (2023), since it was noted that voxel sizes could matter a lot. We model the 3D world in the $[-80\text{m}, 80\text{m}] \times [-80\text{m}, 80\text{m}] \times [-4.5\text{m}, 4.5\text{m}]$ region around the ego vehicle; after initial PointNet (with feature dimension $64$), we obtain a 3D feature volume of tensor shape $1024 \times 1024 \times 64 \times 64$. Following the 3D object detection literature, we pool the 3D feature volume into a 2D Bird-Eye View (BEV) representation, using our aggregation function (sum operation + LayerNorm) on the $z$-axis, after going through another Linear layer and adding a learnable embedding based on the $z$-axis of a voxel.

The encoder backbone is a Swin Transformer (Liu et al., 2021). We add ViT-style (Dosovitskiy et al., 2020) absolute positional encodings of spatial coordinates to the beginning of the backbone.

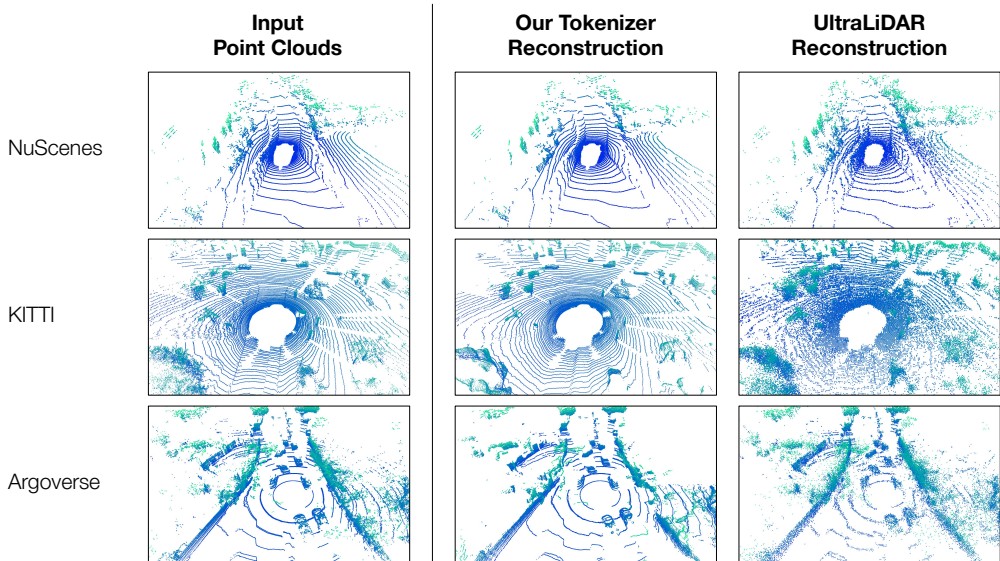

Figure 9: **Qualitative comparison between Copilot4D tokenizer and UltraLiDAR** (Xiong et al., 2023). The UltraLiDAR model also uses a VQ-VAE for point clouds, but it only predicts whether a voxel has points present or not, and is unable to reconstruct more fine-grained geometry. Our tokenizer overcomes this challenge via a combination of implicit representation and differentiable depth rendering, leading to significantly improved reconstructions and qualitatively different results.

The initial patch size is $4$ for the first two Swin layers (with the feature dimension being $128$, and the number of attention heads being $8$), leading to a feature map shape of $256 \times 256 \times 128$. We then downsample the feature map and increase the patch size from $4$ to $8$ via a Patch Merging layer, and apply 6 Swin Transformer layers (with the feature dimension being $256$, and the number of attention heads being $16$), leading to a feature map shape of $128 \times 128 \times 256$. The resolution of the encoder output is $128 \times 128$, which is $8\times$ downsampled from the initial voxel size, meaning that each feature map grid is in charge of a $1.25\text{m} \times 1.25\text{m}$ region in BEV. We find it important to place a LayerNorm (followed by GELU activation (Hendrycks & Gimpel, 2016) and a Linear layer) on the encoder output before vector quantization, which has also been observed in Huh et al. (2023). Following Xiong et al. (2023), we increase the feature dimension to $1024$ with a linear layer before vector quantization; we also find this to be important for good reconstruction.

**Vector quantization layer**  Codebook collapse, meaning that only a few codes are used, is known to be a common issue in VQVAE training. We empirically find that the random restart strategy proposed in Dhariwal et al. (2020) is insufficient for avoiding codebook collapse in our case. We instead resort to the K-Means clustering strategy in Xiong et al. (2023). More specifically, we set up a memory bank to store the most recent encoder outputs before vector quantization; the size of the memory bank is $10$ times the size of the codebook. We define a code to be a *dead code* if it has not been used for 256 iterations. If more than $3\%$ of the entire codebook have become dead codes, then we run K-Means clustering on the memory bank to re-intialize the entire codebook. However, each codebook must go through at least 200 iterations before it can be re-initialized.

We use the straight-through gradient estimator (Bengio et al., 2013), as done in the original VQVAE. In the vector quantization loss $\mathcal{L}_{\text{vq}} = \lambda_1 \|\text{sg}[E(\mathbf{o})] - \hat{\mathbf{z}}\|_2^2 + \lambda_2 \|\text{sg}[\hat{\mathbf{z}}] - E(\mathbf{o})\|_2^2$, we set $\lambda_1 = 0.25$ and $\lambda_2 = 1.0$; the intuition is that the codebook should change more slowly than the features.

**Tokenizer decoder**  The decoder backbone is also a Swin Transformer, symmetrical to the encoder backbone. The Patch Merging layer for downsampling in the encoder corresponds to a Patch Upsample layer for upsampling in the decoder: first, we use a linear layer to upsample the feature map (similar to a deconvolution layer); then, apply LayerNorm on each upsampled feature; finally, use another Linear layer to reduce the feature dimension accordingly. The output of the decoder backbone is a $256 \times 256 \times 128$ tensor in 2D BEV, which is then fed into two separate branches.

As mentioned in the main paper, the first branch is a 3D neural feature grid (NFG) that supports bilinear interpolation for querying continuous coordinates. The NFG branch uses a LayerNorm and a $\text{Linear}(128, 2^2 \times 64 \times 16)$ layer to get a 3D feature volume of tensor shape $512 \times 512 \times 64 \times 16$, with each voxel represented by a 16-dimensional vector. Once this 3D feature volume has been computed, each query $(x_i, y_i, z_i)$ along different rays only requires bilinear interpolation on this feature volume to obtain an initial 16-dimensional vector, which can then be passed into a lightweight two-layer MLP (we set its hidden dimension to be 32, with ReLU activation) to produce an occupancy value with a sigmoid gate at the end. This occupancy value can then be used for differentiable depth rendering in Equation (4). An L1 loss is applied to supervise the rendered depth, with an additional term that encourages the learned sample weight distribution $w_i$ to concentrate within $\epsilon$ meters of the surface, as outlined in Equation (5). We set the margin $\epsilon = 0.4$.

The second branch is a coarse reconstruction branch that is used for spatial skipping during inference. Given the 2D BEV feature map of tensor shape $256 \times 256 \times 128$ from the decoder backbone, we apply a LayerNorm on each feature, followed by a $\text{Linear}(128, 4^2 \times 64)$ layer, to produce a 3D volume of binary classification logits $1024 \times 1024 \times 64$, which is trained to estimate whether each voxel has points present in the input observation. The bias of the final logit is initialized to be $-5.0$, since most voxels are empty in point cloud observations.

At inference, given discrete BEV tokens and query rays, the tokenizer decoder takes in the BEV tokens to produce the 3D NFG and the coarse binary voxel predictions from the two branches. At first, the spatial skipping branch can provide a coarse estimate of where to sample the points along the rays. This is achieved by adding Logistic noise to the binary logits (Maddison et al., 2016) and then thresholding the binary probabilities. We then apply max pooling on this binary 3D volume to produce a much coarser estimate of binary voxel predictions and consequently increase recall (in our case, we chose to use a max pooling factor of 8 in Bird-Eye View). For a query ray, we search among all intersecting coarse voxels and find all of the coarse voxels estimated to have points present. Subsequently, following standard spatial skipping (Li et al., 2023), for each ray, we only sample points within those coarse voxel grids to query the 3D NFG, leading to much more efficient sampling. The final point clouds are obtained via depth rendering on the NFG branch.

### A.2.2 WORLD MODEL

The world model follows a U-Net (Ronneberger et al., 2015) structure with three feature levels corresponding to $1\times$, $2\times$, and $4\times$ downsampled resolutions. The inputs are $128 \times 128$ tokens; the outputs are $128 \times 128 \times 1024$-dimensional logits where 1024 is the vocabulary size of our VQVAE codebook. Adopting the common practice of language models, we use weight tying (Press & Wolf, 2016) between the embedding layer and the final softmax layer. We use Swin Transformer (Liu et al., 2021) for all spatial attention layers, and GPT-2 blocks (Radford et al., 2019) for all temporal attention layers. Spatial attention and temporal attention are interleaved in the following manner: every 2 spatial blocks are followed by 1 temporal block. The temporal blocks apply attention on same feature map location across frames. The feature dimensions for the three feature levels are $(256, 384, 512)$, with the number of attention heads being $(8, 12, 16)$, meaning that we use a fixed 32-dimension per head across all levels.

**We design our network to be similar to the GPT-2 Transformer**, in the sense that the overall structure of every feature level should be: sum of all residuals in Transformer blocks $\rightarrow$ one final LayerNorm $\rightarrow$ one final Linear layer. The benefit of such a design is that, since the structure of our neural net becomes similar to GPT-2, we can directly apply the initialization and optimization strategies found to work well for language models. To make this design compatible with the U-Net structure (which first transitions down and then transition up to make dense predictions), how to merge information when transitioning up across feature levels becomes important. We use the following architecture:

- The first feature level transitioning down has (2 spatial $\rightarrow$ 1 temporal $\rightarrow$ 2 spatial $\rightarrow$ 1 temporal) blocks, with a window size 8 in spatial blocks. Following Swin Transformer, the Patch Merging layer is used for downsampling.

- The second feature level transitioning down also has (2 spatial $\rightarrow$ 1 temporal $\rightarrow$ 2 spatial $\rightarrow$ 1 temporal) blocks, with a window size 8 in spatial blocks.

- The third feature level transitioning down has (2 spatial → 1 temporal) blocks, with a window size 16 in spatial blocks.

- The network then transitions up to the second feature level using an upsampling layer before adding a residual connection. The goal is to upsample the higher-level feature map and merge it with the lower-level feature map. We name such a layer Level Merging, which borrows designs from the Patch Merging layer: first we use a linear layer to output the $2\times$ upsampled feature map (similar to a deconvolution layer), concatenate with the lower-level feature map, applies LayerNorm on every feature, and uses a linear projection to reduce the feature dimension. A residual connection is then applied.

- Back to the second feature level transitioning up, the feature map goes through (2 spatial → 1 temporal) blocks, followed by another Level Merging layer to return to the first feature level. Predictions are made at the first feature level.

- Back to the first feature level, the feature map goes through additional (2 spatial → 1 temporal → 2 spatial → 1 temporal) blocks.

- We use a final LayerNorm followed by a weight matrix (under weight tying with the initial embedding layer) to output the logits.

All our Transformer layers are *pre-norm* layers, meaning that LayerNorm (Ba et al., 2016) is moved to the input of each sub-block, as done in both GPT-2 and Swin Transformer. In a pre-norm Transformer, the main branch of the network becomes just a sum of the residuals since LayerNorm is only placed inside each residual. Therefore, any information we want to condition the network on can be simply added at the very beginning of the network, and will be processed by all following residual modules. The following inputs are added to the beginning of the Transformer world model:

- The embeddings of given observation tokens (which are discrete indices). The weight of this embedding layer is also used for the final softmax layer. On the input side, masking in done via an additional learnable token, as done in BERT. After the embedding layer, we additonally apply Linear → LayerNorm → Linear.

- The ego vehicle poses, which are the *actions* of the ego vehicle. Those $4 \times 4$ matrices are flattened into a 16-dimensional vector, which then goes through Linear → LayerNorm → Linear, and added to all feature map locations of corresponding temporal frames;

- ViT-style (Dosovitskiy et al., 2020) absolute positional encodings of spatial coordinates, which are the same across all temporal frames;

- Learnable temporal positional encodings, broadcast to all feature map locations of corresponding temporal frames;

Other than the spatio-temporal positional encodings, any information we want to condition the neural net on should first go through Linear → LayerNorm → Linear; we have found that applying LayerNorm on the inputs before any Transformer layers is important for learning stability.

We remove the bias parameter in all Linear layers (Touvron et al., 2023), except for the Linear layers outputting query, key, and value during attention in Swin Transformer blocks.

### A.3 TRAINING TRANSFORMERS

**Initialization** Transformer initialization has long been known to be important for successful optimization. We adopt the initialization and optimization strategies found to work well for language models. Following MT-NLG (Smith et al., 2022), all weights are initialized using *fan-in* initialization with a normal distribution of $0$ mean and a standard deviation of $\sqrt{1/(3H)}$, where $H$ is the input feature dimension of each layer. We note that this initialization strategy is the closest to the GPT-1 (Radford et al., 2018) initialization scheme. In addition, we use residual scaling at initialization as done in GPT-2: on each feature level, we count the number of residual connections $L$ (which is the number of Transformer blocks times 2), and rescale the weight matrix of each linear layer before the residual connection by $\sqrt{1/L}$.

**Optimization** The learning rate schedule has a linear warmup followed by cosine decay (with the minimum of the cosine decay set to be 10% of the peak learning rate), and the $\beta_2$ of AdamW

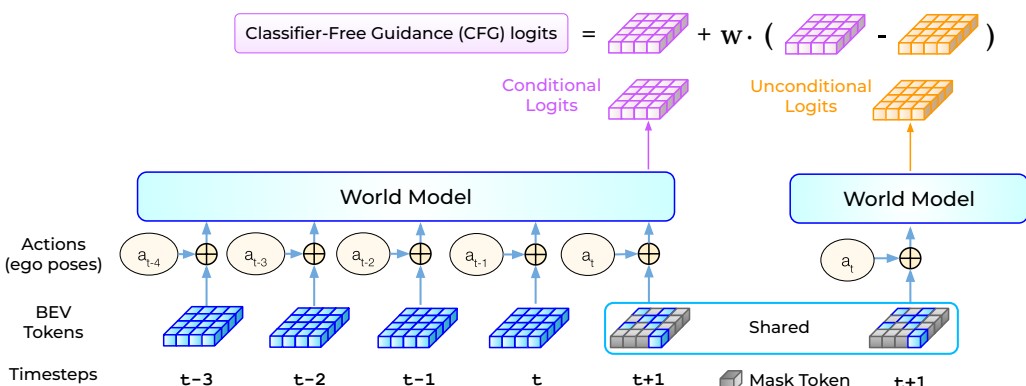

Figure 10: **Illustration of classifier-free diffusion guidance (CFG)** during inference for our world model. The *conditional logits* are conditioned upon the past agent history (past observations and actions). The *unconditional logits* are not conditioned on the past. Classifier-free diffusion guidance amplifies the difference between the two to produce the CFG logits used for sampling. Note that CFG can be efficiently implemented with a single forward pass at each diffusion step by increasing temporal sequence length by 1, and setting the attention mask to be a causal mask within the previous sequence length and an identity mask for the last frame. In addition, thanks to causal masking and our factorized spatio-temporal Transformer, for all past timesteps, we only need cached keys and values from the temporal blocks.

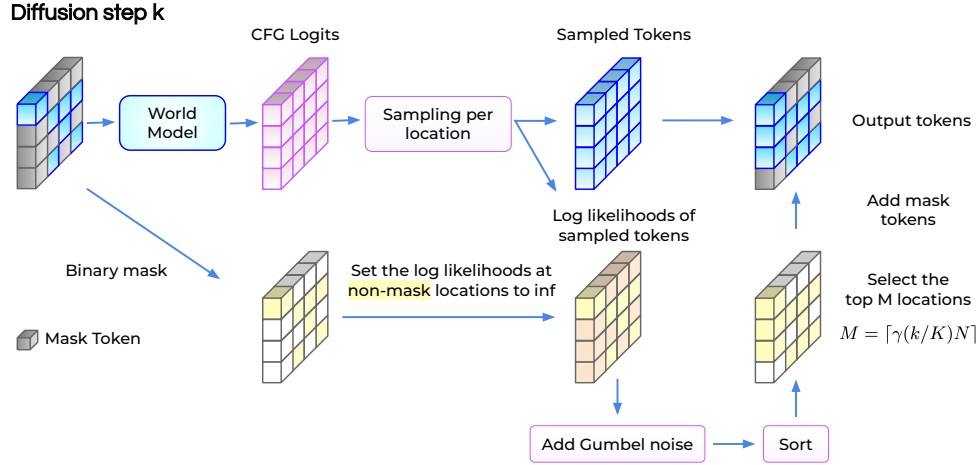

Figure 11: **Illutration of discrete diffusion sampling at each diffusion step**. This procedure corresponds to Algorithm 2. $k$ is the current diffusion step, $K$ is the total number of diffusion steps, $N$ is the number of tokens in each observation, $\gamma(u) = \cos(u\pi/2)$ is the mask schedule.

(Loshchilov & Hutter, 2017) is lowered to be $0.95$. A weight decay of $0.0001$ is applied, but any bias parameters, embedding or un-embedding layers, and LayerNorm parameters are excluded from weight decay. For training tokenizers, we use a learning rate of $0.001$, a linear warmup length of $4000$ iterations, a max gradient norm clipping of $0.1$, a batch size of $16$, and a cosine decay length of $0.4$ million iterations. For training the world model, we use a learning rate of $0.001$, a linear warmup length of $2000$ iterations, a max gradient norm clipping of $5.0$, a batch size of $8$, and a cosine decay length of $0.75$ million iterations. The cross entropy loss uses $0.1$ label smoothing. The same set of hyperparameters are used across all datasets.

### A.4 ALTERNATIVE DERIVATION FOR DISCRETE DIFFUSION TRAINING OBJECTIVE

In discrete diffusion, the forward diffusion process is (Austin et al., 2021)

$$q(x_k \mid x_{k-1}) = \text{Cat}(x_k; p = x_{k-1}Q_k)$$

where Cat refers to a categorical distribution, and $Q_k$ is the forward transition matrix, $[Q_k]_{ij} = q(x_k = j \mid x_{k-1} = i)$ (meaning that each row of $Q$ sums to 1, but not necessarily the columns).

The cumulative transition matrix is

$$\overline{Q}_k = Q_1 Q_2 \cdots Q_k$$

And as a result, if $\overline{Q}_k$ can be written in closed-form, then $q(x_k \mid x_0)$ can be written in terms of $\overline{Q}_k$.

$$q(x_k \mid x_0) = \text{Cat}(x_k; p = x_0\overline{Q}_k)$$

We aim to learn a neural net parameterized by $\theta$ to reverse the forward diffusion process:

$$p_\theta(x_{k-1} \mid x_k) = \sum_{\tilde{x}_0} q(x_{k-1} \mid x_k, \tilde{x}_0)\tilde{p}_\theta(\tilde{x}_0 \mid x_k)$$

**We present an alternative derivation for the lower bound** that we optimize in this paper:

$$\mathbb{E}_{q(x_0)}[\log p_\theta(x_0)]$$

$$= \mathbb{E}_{q(x_0)}[\log \int p_\theta(x_0, x_1 \cdots x_K)\mathrm{d}x_1 \cdots x_K]$$

$$= \mathbb{E}_{q(x_0)}\left\{ \log \mathbb{E}_{q(x_{1:K}|x_0)}\left[ \frac{p_\theta(x_{0:K-1} \mid x_K)}{q(x_{1:K} \mid x_0)}p(x_K) \right]\right\}$$

$$\geq \mathbb{E}_{q(x_0)q(x_{1:K}|x_0)}\left[ \log \frac{p_\theta(x_{0:K-1} \mid x_K)}{q(x_{1:K} \mid x_0)} + \log p(x_K) \right]$$

$$= \mathbb{E}_{q(x_{0:K})}\left[ \sum_{k\geq 1}^{K} \log \frac{p_\theta(x_{k-1} \mid x_k)}{q(x_k \mid x_{k-1})} + \log p(x_K) \right]$$

$$= \mathbb{E}_{q(x_{0:K})}\left[ \sum_{k\geq 1}^{K} \log p_\theta(x_{k-1} \mid x_k) + \log p(x_K) - \sum_{k\geq 1}^{K} \log q(x_k \mid x_{k-1}) \right]$$

$$= \mathbb{E}_{q(x_{0:K})}\left[ \sum_{k\geq 1}^{K} \log \sum_{\tilde{x}_0} q(x_{k-1} \mid x_k, \tilde{x}_0)\tilde{p}_\theta(\tilde{x}_0 \mid x_k) \right] + \underbrace{\mathbb{E}_{q(x_{0:K})}\left[ \log p(x_K) - \sum_{k\geq 1}^{K} \log q(x_k \mid x_{k-1}) \right]}_{C_1}$$

$$= \mathbb{E}_{q(x_{0:K})}\left[ \sum_{k\geq 1}^{K} \log \sum_{\tilde{x}_0} \frac{q(x_{k-1}, \tilde{x}_0 \mid x_k)}{q(\tilde{x}_0 \mid x_k)}\tilde{p}_\theta(\tilde{x}_0 \mid x_k) \right] + C_1$$

$$= \mathbb{E}_{q(x_{0:K})}\left[ \sum_{k\geq 1}^{K} \log \sum_{\tilde{x}_0} \frac{q(\tilde{x}_0 \mid x_{k-1})}{q(\tilde{x}_0 \mid x_k)} \overbrace{q(x_{k-1} \mid x_k)}^{q(x_k|x_{k-1})q(x_{k-1})/q(x_k)} \tilde{p}_\theta(\tilde{x}_0 \mid x_k) \right] + C_1$$

$$\geq \mathbb{E}_{q(x_{0:K})}\left[ \sum_{k\geq 1}^{K} \sum_{\tilde{x}_0} q(\tilde{x}_0 \mid x_{k-1}) \log \left( \frac{q(x_{k-1} \mid x_k)}{q(\tilde{x}_0 \mid x_k)}\tilde{p}_\theta(\tilde{x}_0 \mid x_k) \right) \right] + C_1$$

$$= \mathbb{E}_{q(x_{0:K})}\left[ \sum_{k\geq 1}^{K} \sum_{\tilde{x}_0} q(\tilde{x}_0 \mid x_{k-1}) \log \tilde{p}_\theta(\tilde{x}_0 \mid x_k) \right] + C_1 + \underbrace{\mathbb{E}_{q(x_{0:K})}\left[ \sum_{k\geq 1}^{K} \sum_{\tilde{x}_0} q(\tilde{x}_0 \mid x_{k-1}) \log \frac{q(x_{k-1} \mid x_k)}{q(\tilde{x}_0 \mid x_k)} \right]}_{C_2}$$

$$= \sum_{k \geq 1}^{K} \mathbb{E}_{q(x_0, x_{k-1}, x_k)} \left[ \sum_{\tilde{x}_0} q(\tilde{x}_0 \mid x_{k-1}) \log \tilde{p}_\theta(\tilde{x}_0 \mid x_k) \right] + C_1 + C_2$$

$$= \sum_{k \geq 1}^{K} \mathbb{E}_{q(x_0, x_{k-1}, x_k) q(\tilde{x}_0 \mid x_{k-1})} [\log \tilde{p}_\theta(\tilde{x}_0 \mid x_k)] + C_1 + C_2$$

$$= \sum_{k \geq 1}^{K} \mathbb{E}_{q(x_0 \mid x_{k-1}) q(x_k \mid x_{k-1}) q(x_{k-1}) q(\tilde{x}_0 \mid x_{k-1})} [\log \tilde{p}_\theta(\tilde{x}_0 \mid x_k)] + C_1 + C_2$$

$$= \sum_{k \geq 1}^{K} \mathbb{E}_{q(x_k \mid x_{k-1}) q(x_{k-1}, \tilde{x}_0)} [\log \tilde{p}_\theta(\tilde{x}_0 \mid x_k)] + C_1 + C_2$$

$$= \sum_{k \geq 1}^{K} \mathbb{E}_{q(x_k, x_0)} [\log \tilde{p}_\theta(x_0 \mid x_k)] + C_1 + C_2$$

Analyzing the constants $C_1$ and $C_2$:

$$C_1 = \mathbb{E}_{q(x_{0:K})} \left[ - \sum_{k=1}^{K} \log q(x_k \mid x_{k-1}) + \underbrace{\log p(x_K)}_{\text{Note that } p(x_K) = q(x_K)} \right]$$

$$= \mathbb{E}_{q(x_{0:K})} \left[ - \sum_{k=1}^{K} \log q(x_k, x_{k-1}) + \sum_{k=0}^{K} \log q(x_k) \right]$$

$$C_2 = \mathbb{E}_{q(x_{0:K})} \left[ \sum_{k=1}^{K} \log q(x_{k-1} \mid x_k) \right] - \mathbb{E}_{q(x_{0:K})} \left[ \sum_{k=1}^{K} \sum_{\tilde{x}_0} q(\tilde{x}_0 \mid x_{k-1}) \log q(\tilde{x}_0 \mid x_k) \right]$$

$$= \mathbb{E}_{q(x_{0:K})} \left[ \sum_{k=1}^{K} \log q(x_k, x_{k-1}) - \sum_{k=1}^{K} \log q(x_k) \right] - \sum_{k=1}^{K} \mathbb{E}_{q(x_{0:K}) q(\tilde{x}_0 \mid x_{k-1})} [\log q(\tilde{x}_0 \mid x_k)]$$

$$C_1 + C_2 = \mathbb{E}_{q(x_{0:K})} \left[ \log q(x_0) - \sum_{k=1}^{K} \log q(x_0 \mid x_k) \right]$$

Therefore, this inequality eventually becomes:

$$\mathbb{E}_{q(x_0)} [\log p_\theta(x_0)] \geq \sum_{k=1}^{K} \mathbb{E}_{q(x_k, x_0)} [\log p_\theta(x_0 \mid x_k)] + \mathbb{E}_{q(x_{0:K})} \left[ \log q(x_0) - \sum_{k=1}^{K} \log q(x_0 \mid x_k) \right]$$

$$= \sum_{k=1}^{K} \mathbb{E}_{q(x_0) q(x_k \mid x_0)} [\log p_\theta(x_0 \mid x_k)] + C$$

Which arrives at Equation (6). This loss function avoids the computation of $q(x_{k-1} \mid x_k, x_0)$ as defined in Equation (8) and $p_\theta(x_{k-1} \mid x_k)$ as defined in Equation (1).

## A.5 ABSORBING-UNIFORM DISCRETE DIFFUSION

In this section, we review absorbing-uniform discrete diffusion, and illustrate its connection to our improved version of MaskGIT in detail. First, we aim to express all transition matrices in terms of matrix exponential, which provides a convenient expression for the *cumulative* transtion matrix:

$$
Q_k = \exp(\lambda_k \boldsymbol{R}) = \sum_{n=0}^{\infty} \frac{\lambda_k^n}{n!} \boldsymbol{R}^n
$$

$$
\overline{Q}_k = \exp\left(\left(\sum_{s \leq k} \lambda_s\right)\boldsymbol{R}\right)
$$

In **absorbing diffusion**, each token has a $\alpha_k$ probability of turning into a mask token at step $k$ (using a slightly overloaded notation, not to be confused with the occupancy values in the depth rendering equation). Let $\boldsymbol{e}_m$ be a one-hot vector where the index of the mask token is 1.

$$
\begin{aligned}
Q_k^a = \exp(\gamma_k \boldsymbol{R}_a) &= \exp(\gamma_k(\mathbb{1}\boldsymbol{e}_m^\top - I)) \\
&= \exp(-\gamma_k)I + (1 - \exp(-\gamma_k))\mathbb{1}\boldsymbol{e}_m^\top \\
&= (1 - \alpha_k)I + \alpha_k \mathbb{1}\boldsymbol{e}_m^\top
\end{aligned}
$$

Which utilizes the fact that $(\mathbb{1}\boldsymbol{e}_m^\top - I)(\mathbb{1}\boldsymbol{e}_m^\top - I) = (-1)(\mathbb{1}\boldsymbol{e}_m^\top - I)$ and therefore $(\mathbb{1}\boldsymbol{e}_m^\top - I)^n = (-1)^{n+1}(\mathbb{1}\boldsymbol{e}_m^\top - I)$. Breaking down cumulative absorbing transition matrix $\overline{Q}_k^a$:

$$
\begin{aligned}
\overline{Q}_k^a &= \exp\left(\left(\sum_{s \leq k} \gamma_s\right)(\mathbb{1}\boldsymbol{e}_m^\top - I)\right) \\
&= \exp\left(-\sum_{s \leq k} \gamma_s\right)I + \left(1 - \exp\left(-\sum_{s \leq k} \gamma_s\right)\right)\mathbb{1}\boldsymbol{e}_m^\top \\
&= \prod_{s \leq k}(1 - \alpha_s)I + \left(1 - \prod_{s \leq k}(1 - \alpha_s)\right)\mathbb{1}\boldsymbol{e}_m^\top
\end{aligned}
$$

In **uniform diffusion**, each token has a $\beta_k$ probability of turning into a random non-mask token at step $k$. With $M$ non-mask categories ($M = |V|$), the transition matrix can be expressed as:

$$
\begin{aligned}
Q_k^u &= I - \beta_k(I - \boldsymbol{e}_m\boldsymbol{e}_m^\top) + \frac{\beta_k}{M}(\mathbb{1} - \boldsymbol{e}_m)(\mathbb{1} - \boldsymbol{e}_m)^\top \\
&= (1 - \beta_k)I + \beta_k\left(\boldsymbol{e}_m\boldsymbol{e}_m^\top + \frac{1}{M}(\mathbb{1} - \boldsymbol{e}_m)(\mathbb{1} - \boldsymbol{e}_m)^\top\right) \\
&= \exp\left(\eta_k \boldsymbol{R}_u\right) = \exp\left(\eta_k\left[\frac{1}{M}(\mathbb{1} - \boldsymbol{e}_m)(\mathbb{1} - \boldsymbol{e}_m)^\top - (I - \boldsymbol{e}_m\boldsymbol{e}_m^\top)\right]\right) \\
&= \exp(-\eta_k)I + (1 - \exp(-\eta_k))\left(\boldsymbol{e}_m\boldsymbol{e}_m^\top + \frac{1}{M}(\mathbb{1} - \boldsymbol{e}_m)(\mathbb{1} - \boldsymbol{e}_m)^\top\right)
\end{aligned}
$$

Breaking down cumulative uniform transition matrix $\overline{Q}_k^u$:

$$
\begin{aligned}
\overline{Q}_k^u &= \exp\left(\left(\sum_{s \leq k} \eta_s\right)\left[\frac{1}{M}(\mathbb{1} - \boldsymbol{e}_m)(\mathbb{1} - \boldsymbol{e}_m)^\top - (I - \boldsymbol{e}_m\boldsymbol{e}_m^\top)\right]\right) \\
&= \exp\left(-\sum_{s \leq k} \eta_s\right)I + \left(1 - \exp\left(-\sum_{s \leq k} \eta_s\right)\right)\left(\boldsymbol{e}_m\boldsymbol{e}_m^\top + \frac{1}{M}(\mathbb{1} - \boldsymbol{e}_m)(\mathbb{1} - \boldsymbol{e}_m)^\top\right) \\
&= \prod_{s \leq k}(1 - \beta_s)I + \left(1 - \prod_{s \leq k}(1 - \beta_s)\right)\left(\boldsymbol{e}_m\boldsymbol{e}_m^\top + \frac{1}{M}(\mathbb{1} - \boldsymbol{e}_m)(\mathbb{1} - \boldsymbol{e}_m)^\top\right)
\end{aligned}
$$

In **absorbing-uniform diffusion**, with $e_m^\top$ being the $(M+1)$-th row, the transition matrix is:

$$Q_k = Q_k^a Q_k^u$$

$$= \begin{bmatrix} \omega_k + \nu_k & \nu_k & \nu_k & \cdots & \nu_k & \alpha_k \\ \nu_k & \omega_k + \nu_k & \nu_k & \cdots & \nu_k & \alpha_k \\ \nu_k & \nu_k & \omega_k + \nu_k & \cdots & \nu_k & \alpha_k \\ \vdots & \vdots & \vdots & \ddots & \vdots & \vdots \\ \nu_k & \nu_k & \nu_k & \cdots & \omega_k + \nu_k & \alpha_k \\ 0 & 0 & 0 & \cdots & 0 & 1 \end{bmatrix}$$

$$\nu_k = \frac{\beta_k}{M} \quad \omega_k = 1 - \beta_k - \alpha_k$$

Note that $Q_k^a$ and $Q_k^u$ are commuting matrices; $\boldsymbol{R}_a$ and $\boldsymbol{R}_u$ inside matrix exponentials also commute (so that $\exp(\eta_k \boldsymbol{R}_u) \exp(\gamma_k \boldsymbol{R}_a) = \exp(\eta_k \boldsymbol{R}_u + \gamma_k \boldsymbol{R}_a)$). This property allows us to write the cumulative transition matrix as:

$$\overline{Q}_k = \overline{Q}_k^a \overline{Q}_k^u$$

$$= \begin{bmatrix} \overline{\omega}_k + \overline{\nu}_k & \overline{\nu}_k & \overline{\nu}_k & \cdots & \overline{\nu}_k & \overline{\chi}_k \\ \overline{\nu}_k & \overline{\omega}_k + \overline{\nu}_k & \overline{\nu}_k & \cdots & \overline{\nu}_k & \overline{\chi}_k \\ \overline{\nu}_k & \overline{\nu}_k & \overline{\omega}_k + \overline{\nu}_k & \cdots & \overline{\nu}_k & \overline{\chi}_k \\ \vdots & \vdots & \vdots & \ddots & \vdots & \vdots \\ \overline{\nu}_k & \overline{\nu}_k & \overline{\nu}_k & \cdots & \overline{\omega}_k + \overline{\nu}_k & \overline{\chi}_k \\ 0 & 0 & 0 & \cdots & 0 & 1 \end{bmatrix}$$

$$\overline{\omega}_k = \prod_{s \leq k}(1 - \beta_s - \alpha_s) \quad \overline{\chi}_k = 1 - \prod_{s \leq k}(1 - \alpha_s) \quad \overline{\nu}_k = \frac{1}{M}(1 - \overline{\omega}_k - \overline{\chi}_k)$$

Intuitively, $\overline{\omega}_k$ is the probability that a ground-truth token neither flips into a random token nor gets absorbed into the mask token after $k$ forward diffusion steps. $\overline{\chi}_k$ is the probability of having a mask token after $k$ forward diffusion steps.

Given that the posterior can be expressed as

$$q(x_{k-1} \mid x_k, x_0) = \frac{q(x_k \mid x_{k-1}, x_0)q(x_{k-1} \mid x_0)}{q(x_k \mid x_0)} = \frac{q(x_{k-1} \mid x_0)}{q(x_k \mid x_0)}q(x_k \mid x_{k-1}) \tag{8}$$

We can express it in matrix form for discrete diffusion:

$$q(x_{k-1} = i \mid x_k = j, x_0 = l) = \frac{[\overline{Q}_{k-1}]_{li} \cdot [Q_k]_{ij}}{[\overline{Q}_k]_{lj}} \quad q(x_{k-1} \mid x_k, x_0) = \frac{x_0 \overline{Q}_{k-1} \odot x_k Q_k^\top}{x_0 \overline{Q}_k x_k^\top}$$

Allowing us to efficiently compute the closed-form posterior from $\overline{Q}_{k-1}$, $\overline{Q}_k$, and $Q_k$ defined earlier.

## A.6 TRAINING AND INFERENCE OF ABSORBING-UNIFORM DISCRETE DIFFUSION

**During training**, we first apply masking, which corresponds to absorbing diffusion with $\overline{Q}_k^a$, and then add uniform noise, which corresponds to uniform diffusion with $\overline{Q}_k^u$. We add at a maximum $\eta\%$ of uniform noise under a linear schedule, where we set $\eta = 20$ by default. $\eta$ controls the minimum signal-to-noise ratio among the non-mask tokens.

- Despite its simplicity, Algorithm 1 is training an absorbing-uniform discrete diffusion model. To see this, we simply need to define the corresponding noise schedules for uniform diffusion and absorbing diffusion. The uniform discrete diffusion noise schedule is set to be $\beta_k = 1/(K/\eta - k + 1)$ and consequently $(1 - \prod_{s \leq k}(1 - \beta_s)) = \eta \cdot k/K$. Since $(1 - \prod_{s \leq k}(1 - \beta_s))$ has now become a linear function with respect to $k$, we can simply uniformly sample a noise ratio without computing any transition matrix, as done in Algorithm 1. As for the absorbing diffusion noise schedule, we define $\overline{\alpha}_k = \prod_{s \leq k}(1 - \alpha_s)$ as a cosine schedule, and $\alpha_k$ can be solved accordingly with $\alpha_k = 1 - \overline{\alpha}_k/\overline{\alpha}_{k-1}$.

**During inference**, we find that when decoding $x_k$, marginalizing over $x_0$ as done in $p_\theta(x_k|x_{k+1}) = \sum_{x_0} q(x_k \mid x_{k+1}, x_0) p_\theta(x_0 \mid x_{k+1})$ is not necessary; instead, we directly decode the predicted $x_0$ from $p_\theta(x_0|x_{k+1})$, as done in MaskGIT. During sampling, once a token has transitioned from a mask token to a non-mask token, even though it can be further revised and resampled, it cannot go back to become masked again, as one would expect from the reverse process of absorbing-uniform diffusion. This is achieved by setting $l_k \leftarrow +\infty$ on non-mask indices of $x_{k+1}$ in Algorithm 2.

We find that, during the sampling process $\tilde{x}_0 \sim p_\theta(\cdot \mid x_{k+1})$, the sampling quality can be improved by using top-$K$ sampling rather than vanilla categorical sampling (similar to what we would expect from a language model). In our experiments, we sample from the top 3 logits per feature location.

## A.7 ADDITIONAL QUALITATIVE RESULTS

We present additional qualitative results: Figure 13 and Figure 14 outline our results on NuScenes; Figure 15 shows our results on KITTI, and Figure 16 and Figure 17 adds to the results on Argoverse 2 Lidar. Overall, our models achieve considerably better qualitative results on all three datasets.

**Predictions under counterfactual actions**   We visualize how well the world model can predict the future under counterfactual actions in Figure 12. Here we modify the future trajectories of the ego vehicle (which are action inputs to the world model), and we demonstrate that the world model is able to imagine alternative futures given different actions. The imagined futures are consistent with the counterfactual action inputs. Moreover, the world model has learned that other vehicles in the scene are reactive agents.

**Failure cases of our current models**   We present some failure modes of our current models using red arrows in Figure 14, Figure 16, and Figure 17. Modeling vehicle behaviours on a 3s time horizon needs further improvement, as the accuracy for 3s prediction seems lower than 1s prediction; however, we note that this inaccuracy could be partly due to the multi-modality of 3s future prediction (multiple futures could be equally valid). Additionally, on a 3s time horizon, sometimes new vehicles (not present in the past or present frames) enter the scene. Currently, the world model does not seem to have learned how to hallucinate new vehicles coming into the region of interest. Nevertheless, those failure cases do not necessarily reflect fundamental limitations of our proposed algorithm; we believe that the issues listed above can be addressed to a large extent by increasing data, compute, and model size under our framework.

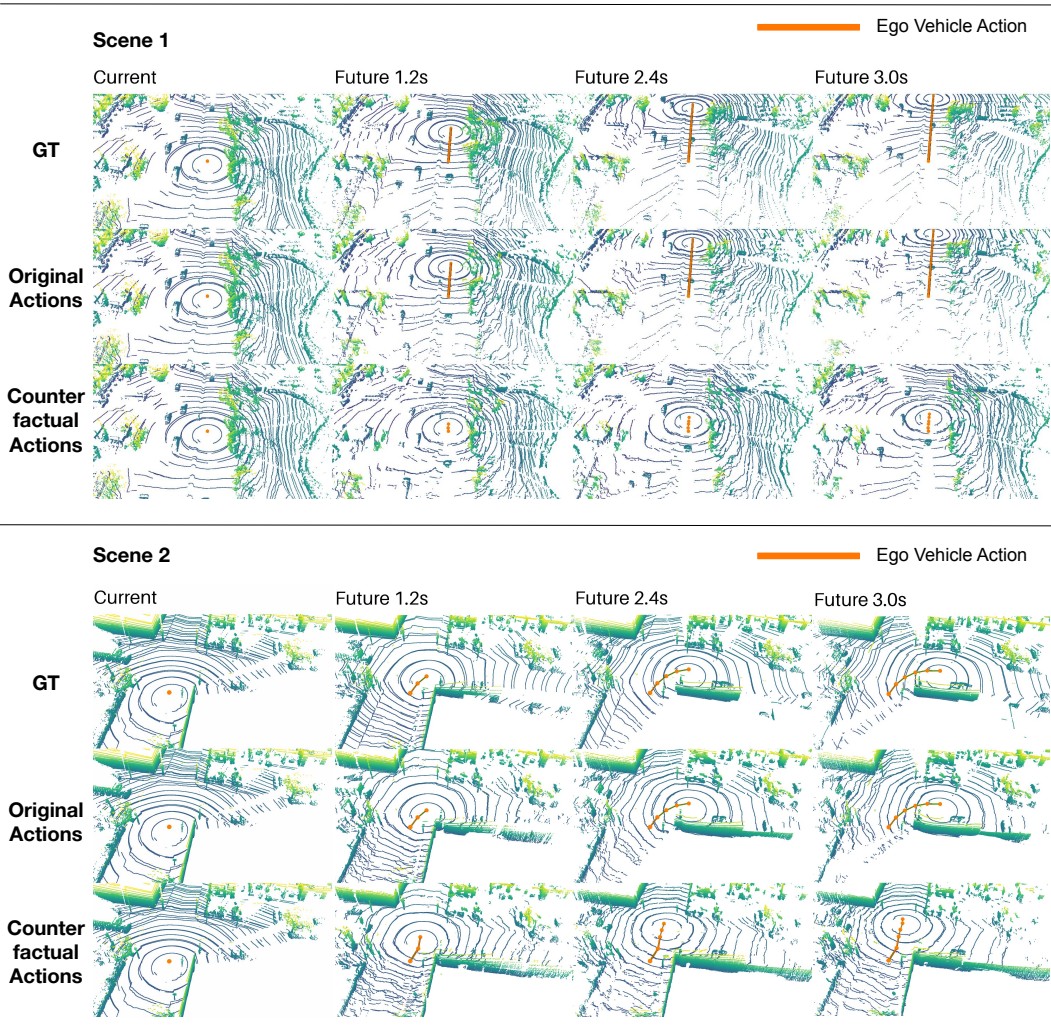

Figure 12: **Visualization of predicted future under counterfactual actions.** Here we modify the future trajectories of the ego vehicle (which are action inputs to the world model), and demonstrate that the world model is able to imagine alternative futures given different actions. The orange trajectory represents the actions taken by the ego vehicle regarding where to drive. The imagined futures are consistent with the counterfactual action inputs. In addition, our world model is able to learn that other vehicles in the scene are reactive agents. Notably, in Scene 1, the counterfactual action is for the ego vehicle to brake, and our world model has learned that the vehicle behind will also react by braking to avoid a collision with the ego vehicle.

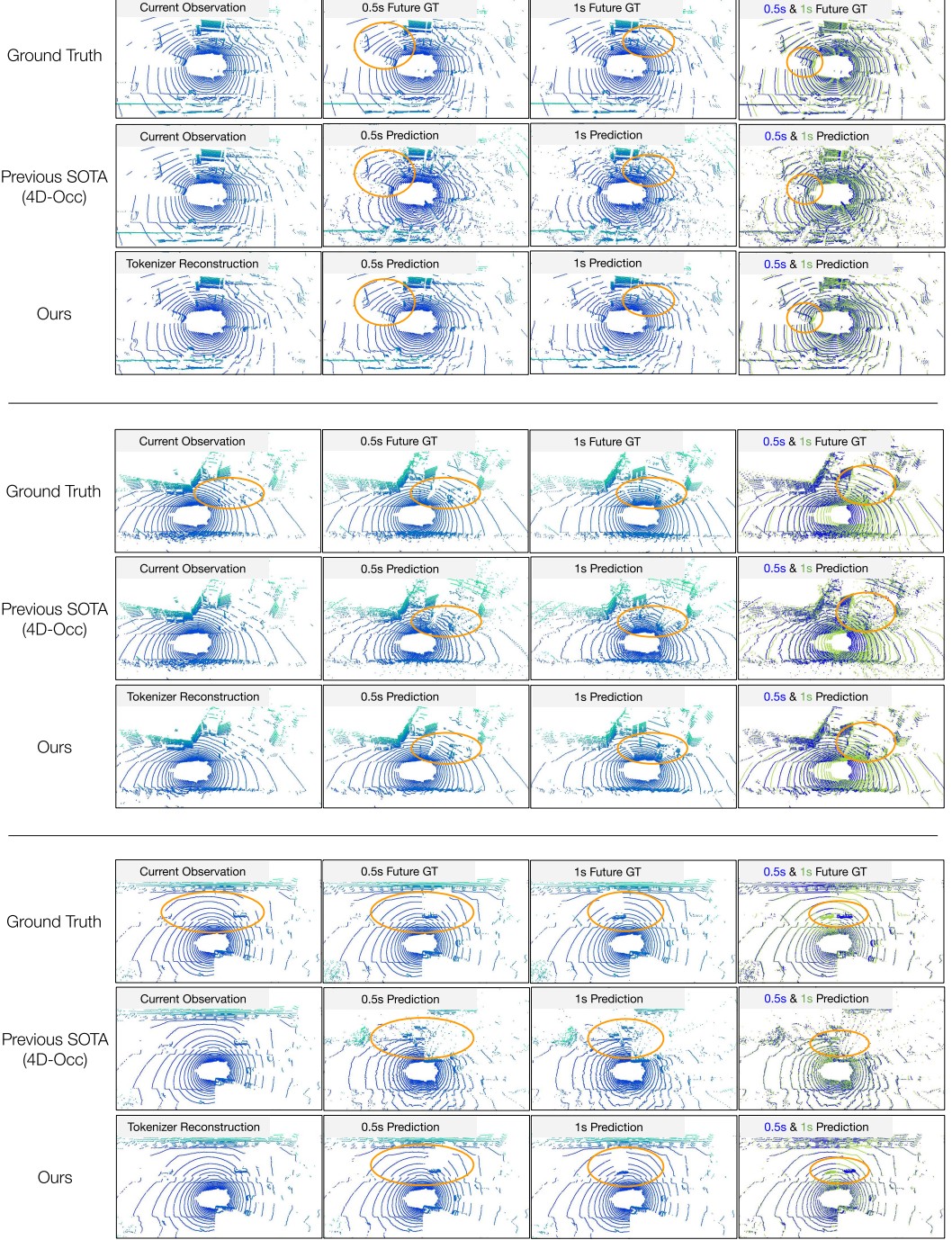

Figure 13: **Qualitative comparison on NuScenes 1s prediction**. The last column overlaps the point clouds from 0.5s prediction and 1s prediction to make clear how vehicles are moving.

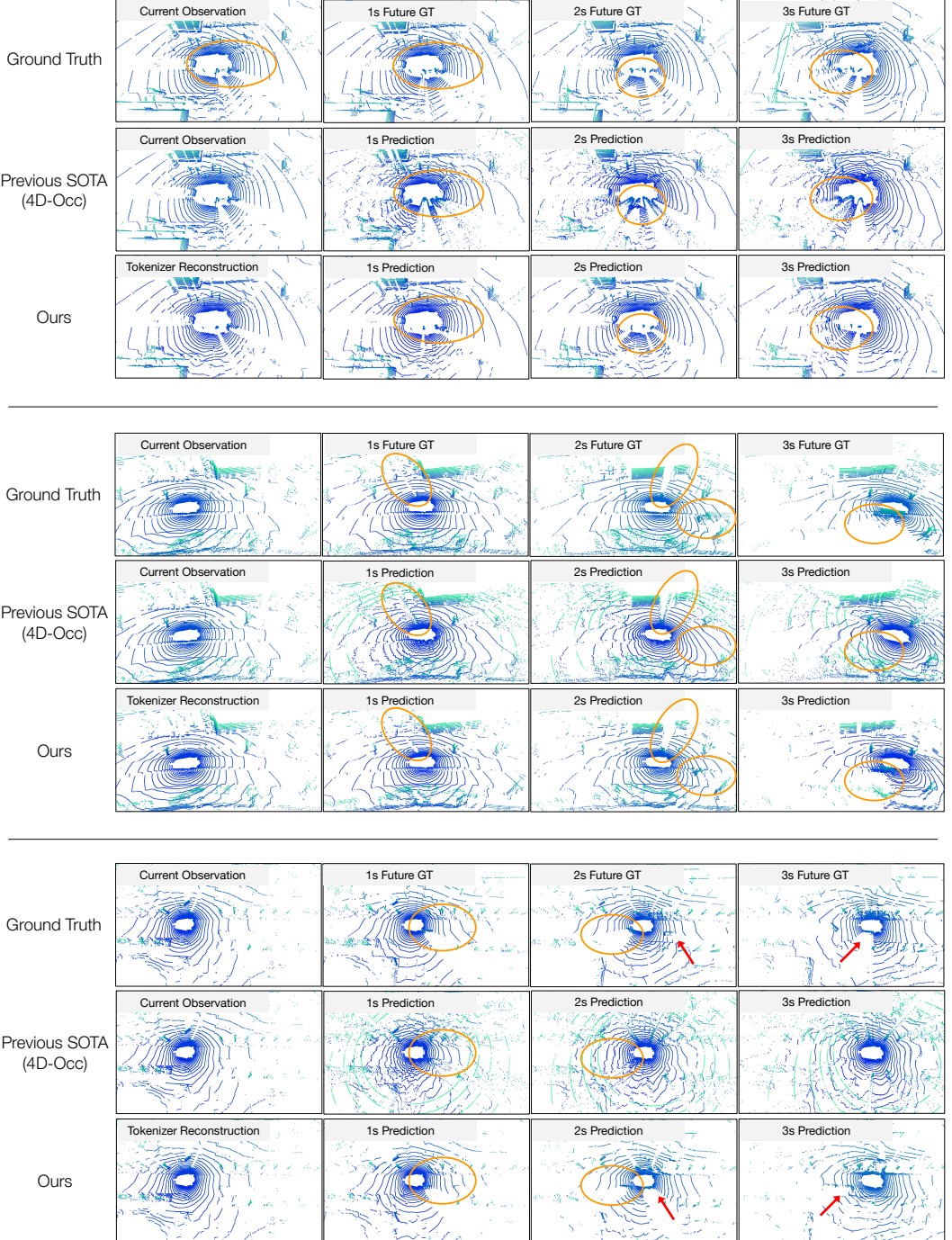

Figure 14: **Qualitative comparison on NuScenes 3s prediction**. The orange circles highlight where our model does well; the red arrows point out some failure cases of our model.

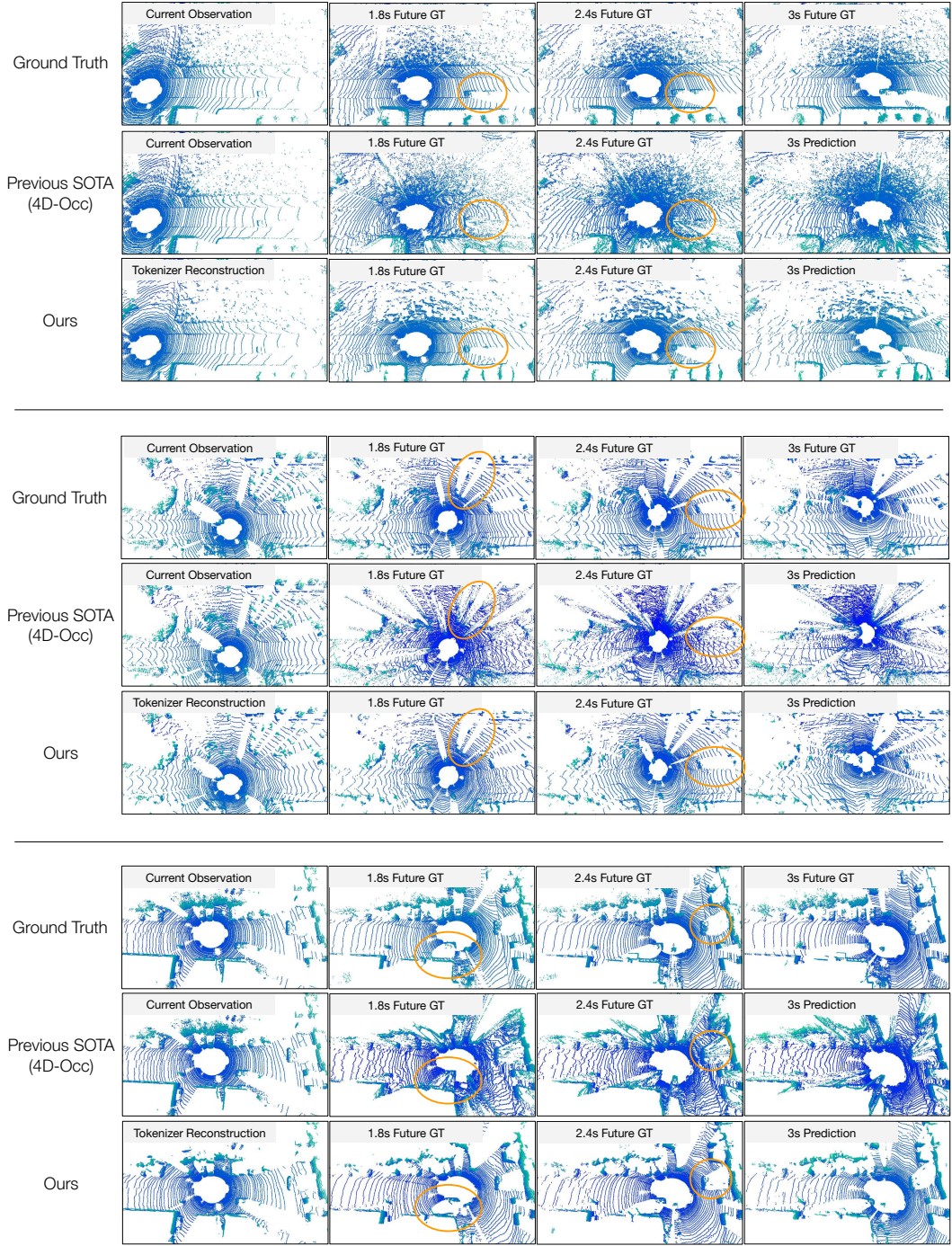

Figure 15: **Qualitative comparison on KITTI Odometry**. Note that the color of a point is merely dependent on its height ($z$-axis value); therefore, if the colors of the predicted points are different from the colors of the ground-truth points, it means that the predicted point heights are off. Our method clearly achieves significantly better qualitative results compared to prior SOTA.

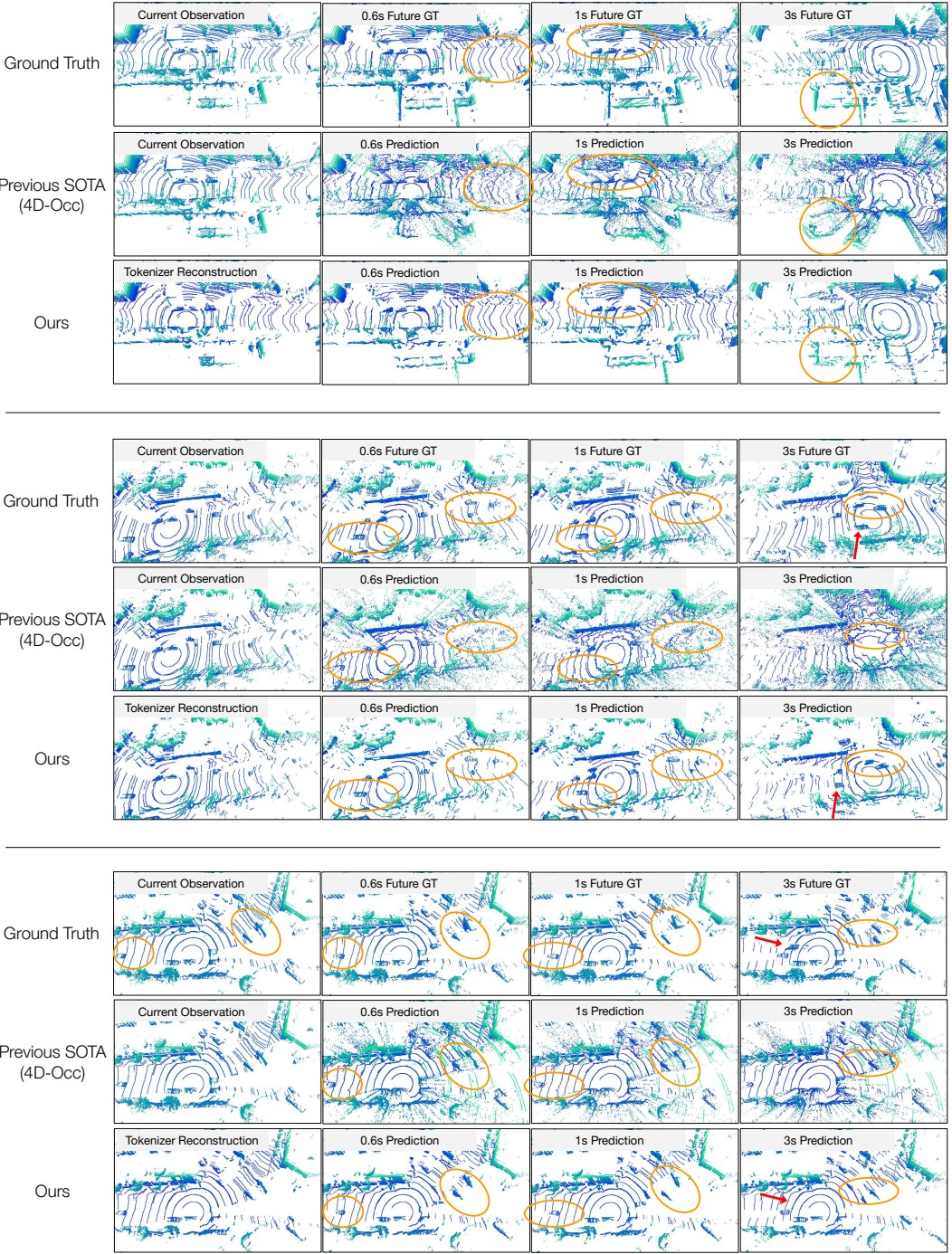

Figure 16: **Additional qualitative comparison on Argoverse2 Lidar dataset**. The orange circles highlight where our model does well; the red arrows point out some failure cases of our model.

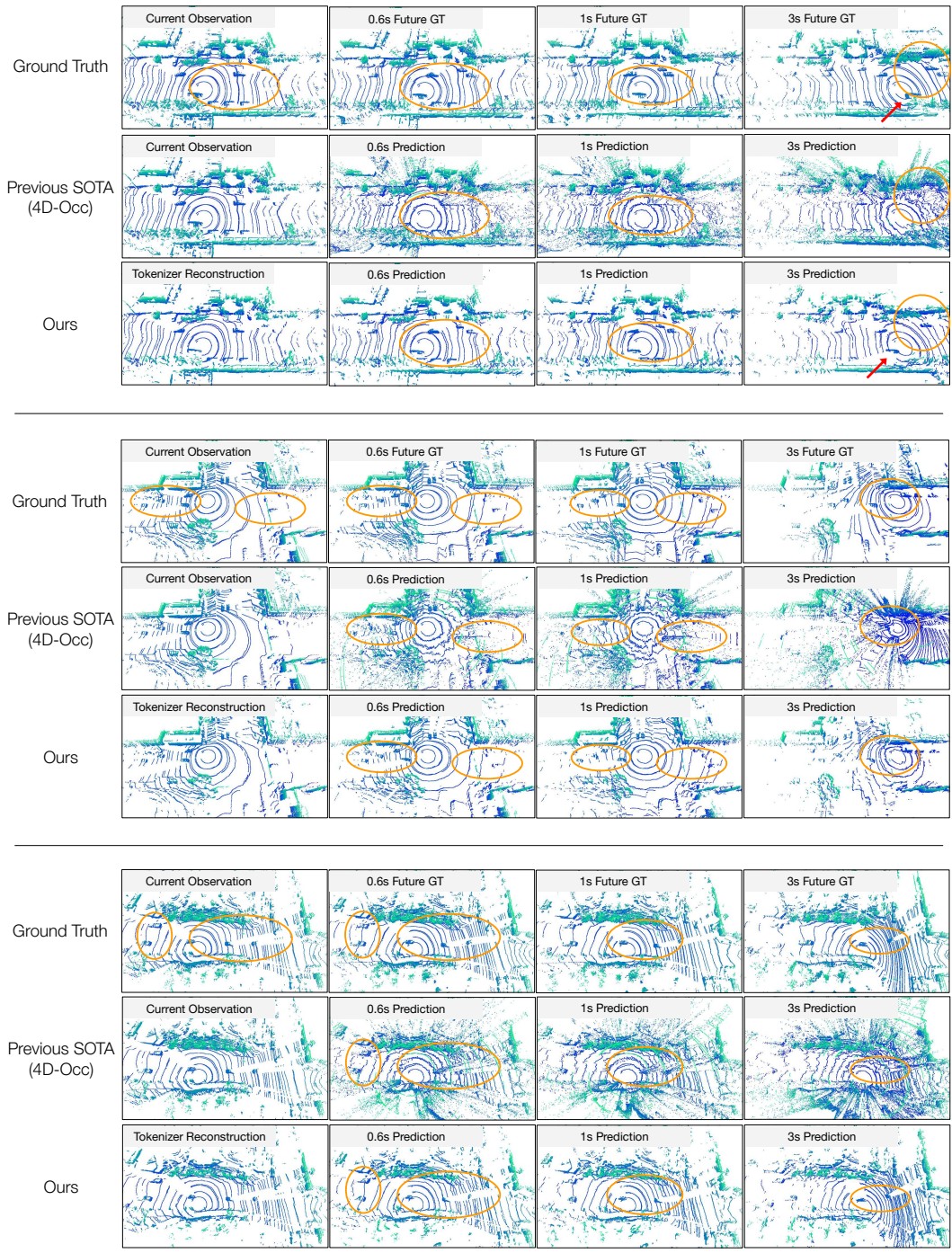

Figure 17: **Additional qualitative comparison on Argoverse2 Lidar dataset**. The orange circles highlight where our model does well; the red arrows point out some failure cases of our model.

