# OpenReview forum: "Copilot4D: Learning Unsupervised World Models for Autonomous Driving via Discrete Diffusion"
_ICLR.cc/2024/Conference — ICLR 2024 poster_

### Official Review · Reviewer_4daR · 2023-10-30

**Soundness:** 3 good
**Presentation:** 3 good
**Contribution:** 3 good
**Rating:** 6
**Confidence:** 4

**Summary:**

This paper proposes a point cloud forecasting-based world model for autonomous driving. The model first tokenizes point clouds into discrete BEV tokens (codebook/vocabulary) following VQVAE and UltraLiDAR (Xiong et al., 2023). Then tokens are decoded to reconstruct the point clouds with an implicit representation depth rendering branch and a classical coarse voxel reconstruction branch. MaskGIT is further leveraged to a discrete diffusion model, with different masking conditions and history information (the classifier-free diffuision) guidance applied, to realize the future prediction ability. The point cloud forecasting method is evaluated on three datasets and achieves state-of-the-art results.

**Strengths:**

- The presentation of the paper is good, especially in the methodology part. Symbols and figures are clear and helpful for understanding.
- The model architecture is detailed in the appendix. The authors also provide details besides the model structure, such as the K-means clustering strategy to solve codebook collapse and LayerNorm to stabilize training, which are valuable empirical findings for future research.
- MaskGIT with diffusion is interesting. It could probably be applied to other tasks as well.

**Weaknesses:**

- The reviewer is confused about the motivation of discrete tokenization and masked image modeling.
  -  The proposed method adopts a VQVAE-like model to capture the complex 3D world, as mentioned in the introduction challenge (i). Classic BEV (the method in BEVFusion, BEVFormer, etc.,) can also realize this ability IMO. This undermines the motivation to use discrete tokenization and the necessity to use a discrete diffusion model in the after.
  -  Table 4 presents the ablation of the discrete diffusion algorithm. The motivation to use MaskGIT seems its parallel decoding strategy. How about the masked image modeling? What will the results or inference time be like if no masked image modeling (or even MaskGIT) is applied?
  -  The intuition of using different masking strategies for world model training comes from the robotics field. It would be valuable if ablations on this could be presented as well.
  -  In light of the above points, a naive baseline should be simple diffusion modeling with simple BEV features.
- The gain of point cloud forecasting mostly comes from CFG, which involves past poses and actions. Without this, the results are close to 4D-Occ.
  - Taking the current action as input is reasonable for a world model, but much information about history could make the prediction rely heavily on the past. If the prediction horizon is longer, such a long history is also weird as poses at the very beginning are intuitively unhelpful.
  - As this task implicitly involves ego-planning, this could lead to causal confusion though impressive results are obtained under the open-loop scenario. The authors have stated that combining the world modeling approach with model-based RL is a future direction, yet, it is important to demonstrate its effectiveness for the decision-making task.

**Questions:**

- In the introduction, the task definition of point cloud forecasting is 'to predict future point cloud observations given past observations and future ego vehicle poses'. In which paper, the future ego vehicle pose is provided?
- Why not report full results under all metrics in ablation study tables?

---

> ### Author Response · Authors · 2023-11-19
> **Author Response (Part 1)**
>
> We thank the reviewer for their feedback. We are glad that the reviewer finds our approach to be interesting and the presentation of methodology to be clear and helpful. We will be addressing your questions and concerns below.
>
> *Q: “The proposed method adopts a VQVAE-like model to capture the complex 3D world, as mentioned in the introduction challenge (i). Classic BEV (the method in BEVFusion, BEVFormer, etc.,) can also realize this ability IMO.”*
>
> A: Methods like BEVFusion and BEVFormer are image encoders and tackle supervised object detection; they do not have the ability to generate Lidar point clouds.
>
> *Q: “How about the masked image modeling? What will the results or inference time be like if no masked image modeling (or even MaskGIT) is applied?”*
>
> A: Masked image modeling is mostly used for representation learning, and often cannot serve as a generative model on its own. The task of world modeling is a generative task. Regarding the connection between masked image modeling and MaskGIT, it is true that after tokenization, MaskGIT is very similar to masked image modeling, but MaskGIT can serve as a generative model.
>
> *Q: “The intuition of using different masking strategies for world model training comes from the robotics field. It would be valuable if ablations on this could be presented as well.”*
>
> A: We only use two types of attention masks during training, and both are necessary. The masks are either a causal mask, or an identity mask that only allows each frame to attend to itself. The causal attention mask is standard in GPT language models (the original Transformer paper [1] says that they “modify the self-attention sub-layer … to prevent positions from attending to subsequent positions”). The identity (attention) mask is necessary for classifier-free diffusion guidance (CFG). Without it, CFG cannot be applied. To better illustrate how CFG works in our world model and why the second type of mask is necessary, we have included an additional diagram Figure 10 in the Appendix.
>
> *Q: “A naive baseline should be simple diffusion modeling with simple BEV features.”*
>
> A: Simple BEV features from supervised object detectors such as BEVFusion and BEVFormer cannot generate point clouds. In addition, assuming that our proposed tokenizer is used, other diffusion models such as Gaussian diffusion models are not necessarily simpler, and often require many more diffusion steps than our discrete diffusion model (which only requires 10 diffusion steps) in the absence of additional distillation techniques. Besides, our proposed method for classifier-free diffusion guidance in the world modeling context is likely important regardless of the continuous/discrete nature of the diffusion world model.
>
> *Q: “Taking the current action as input is reasonable for a world model, but much information about history could make the prediction rely heavily on the past.”*
>
> A: Since a self-driving environment is a partially observable rather than a fully observable environment, conditioning on both past observations and past actions is a reasonable thing to do. The more information from the past that is provided to the world model, the better it can estimate the current state of the world and make predictions about the future. The Transformer can learn to decide which part of information is most relevant or irrelevant, but it can only learn to do so when the necessary information is available.
>
> *Q: “As this task implicitly involves ego-planning, this could lead to causal confusion though impressive results are obtained under the open-loop scenario.” “In which paper, the future ego vehicle pose is provided?”*
>
> A: For a world model in self-driving scenes, the actions are where the vehicle plans to drive, represented by future ego vehicle poses. The previous state-of-the-art method 4D Occupancy [2] also uses future ego vehicle poses. Since the future ego poses are provided to the world model, the world modeling task alone does not involve ego-planning. We acknowledge that transferring this to a closed-loop setting while mitigating causal confusion is an interesting and open research problem, but the goal of our paper is to provide a more accurate world model that can be learned from unlabeled data, as we believe this is the major current bottleneck.
>
> *Q: “Why not report full results under all metrics in ablation study tables?”*
>
> A: In our updated draft, we have updated the ablation study tables to include all metrics within the ROI.

---

> > ### Author Response · Authors · 2023-11-19
> > **Author Response (Part 2)**
> >
> > *Q: “The authors have stated that combining the world modeling approach with model-based RL is a future direction, yet, it is important to demonstrate its effectiveness for the decision-making task”*
> >
> > A: To apply model-based RL (MBRL) approaches to a domain, a prerequisite is that a reasonably good world model can be learned in this domain. Our proposed model represents a leap in the fidelity of unsupervised world models in the self-driving domain, which unlocks the possibility of doing model-based RL on raw driving data. In other words, our work is a necessary step before MBRL can be applied. Adding MBRL introduces additional complexities and is beyond the scope of one paper; as such, we leave this for future work.
> >
> > *References:*
> >
> > [1] Vaswani, Ashish, et al. "Attention is all you need." Advances in neural information processing systems 30 (2017).
> >
> > [2] Khurana, Tarasha, et al. "Point Cloud Forecasting as a Proxy for 4D Occupancy Forecasting." Proceedings of the IEEE/CVF Conference on Computer Vision and Pattern Recognition. 2023.

---

> ### Comment · Reviewer_4daR · 2023-11-22
> **Post-rebuttal Response from Reviewer**
>
> Thanks for the clarifications. I have also detailedly gone over other reviewers' comments and authors' feedback. I have some remaining concerns below.
>
> - The motivation of discrete tokenization, in my original review. Maybe I didn't clearly state my thoughts. This point is somehow similar to the concern of Reviewer 2kSt. The method mainly includes three main components besides the neural rendering decoder, VQVAE-like BEV tokenizer, MaskGIT-based discrete diffusion model, and the spatio-temporal world model. My original concern is why use VQVAE-like BEV tokenizer (discrete BEV tokens) instead of continuous BEV features like normal BEV methods (BEVFormer, etc) or even voxel-based methods? The point is continuous vs discrete. I cannot fully get the idea of why they do not have the ability to generate Lidar point clouds. They should behave similarly to the tokenized one for the reconstruction decoder.
>   - If thus, an ideal ablation table should be (the last two rows are current Tab.4). They are likely inferior and inefficient, but the comparisons as well as proper motivation in the introduction could make the overall project more solid and convincing:
> | Method         | Metrics    | FPS | Others |
> | -------------- | :-------: | :--------: | :------: |
> | continuous BEV + naive diffusion          |    -   |    -     |   -   |
> | tokenized BEV + naive diffusion       |    -   |    -     |   -   |
> | tokenized BEV + MaskGIT |    -   |    -     |   -   |
> | tokenized BEV + MaskGIT-based discreate diffusion |    -   |    -     |   -   |
>
> - I agree with the authors that conditioning on past observations and actions is reasonable for a world model. My concern is that this could lead to an unfair comparison with previous methods, as suggested by Reviewer amws as well. The gain of point cloud forecasting mostly comes from CFG. Without this, the results are close to 4D-Occ.

---

> > ### Author Response · Authors · 2023-11-23
> > **Author Reply (Follow-up)**
> >
> > We thank the reviewer for the reply and further questions. We will provide further clarifications below.
> >
> > *Q: "My original concern is why use VQVAE-like BEV tokenizer (discrete BEV tokens) instead of continuous BEV features like normal BEV methods (BEVFormer, etc) or even voxel-based methods? The point is continuous vs discrete. I cannot fully get the idea of why they do not have the ability to generate Lidar point clouds."*
> >
> > A: The clarify this issue from the following perspectives:
> >  - What we previously pointed out is that: methods like BEVFormer and BEVFusion that the reviewer initially suggested could not generate point clouds on their own, because they are supervised object detectors. In other words, our proposed tokenizer decoder and tokenizer training objectives are still necessary.
> >  - The suggestions here seems to be: keeping everything in our tokenizer the same and removing the vector quantization layer, and then training a Latent Diffusion Model (LDM) [1] rather than our proposed discrete diffusion model. The LDM paper shows that training such an LDM can be highly non-trivial, and therefore it is not a simple ablation but rather a separate research effort. As mentioned in [1], it is important to "avoid arbitrarily high-variance latent spaces"; even for continuous diffusion in the latent space, latent regularization is still necessary, either through vector quantization (like we did) or a KL-penalty towards a standard normal. In other words, regarding the comparison between "continuous BEV + naive diffusion" and "tokenized BEV + naive diffusion", we already know that regularization on the latent space produces better results from LDM [1].
> >  - Regarding the comparison between continuous versus discrete diffusion (both in the latent space), we point out that MaskGIT-based Muse model [2] has already been shown to outperform the LDM model [1] in generation quality (for text-to-image generation) given the same model size, while only requiring an order of magnitude fewer number of sampling steps at inference.
> >  - Considering that those findings have already been made in other domains of generative modeling, we believe that it is sufficient on our end to cite those papers and their conclusions as a part of our premises, rather than having to rediscover their conclusions in our ablation study.
> >
> > *Q: "I agree with the authors that conditioning on past observations and actions is reasonable for a world model. My concern is that this could lead to an unfair comparison with previous methods, as suggested by Reviewer amws as well."*
> >
> > A: We have explained to Reviewer amws that the past observations and action history are already inputs to the world model; they do not require any additional knowledge, and therefore do not lead to any unfair comparison.
> >
> > *Q: "The gain of point cloud forecasting mostly comes from CFG. Without this, the results are close to 4D-Occ."*
> >
> > A: Discovering that CFG is important for diffusion-based world modeling is in fact a contribution, not a weakness, of our work. Classifier-free diffusion guidance (CFG) is a commonly used technique in text-to-image diffusion modeling, and we figured out a way to apply it to learning world models as well. If anything, it shows that a vanilla application of diffusion to the world modeling context might not have worked very well, and additional innovation was required.
> >
> > References:
> >
> > [1] Rombach, Robin, et al. "High-resolution image synthesis with latent diffusion models." Proceedings of the IEEE/CVF conference on computer vision and pattern recognition. 2022.
> >
> > [2] Chang, Huiwen, et al. "Muse: Text-to-image generation via masked generative transformers." arXiv preprint arXiv:2301.00704 (2023).

---

### Official Review · Reviewer_amws · 2023-10-30

**Soundness:** 3 good
**Presentation:** 3 good
**Contribution:** 4 excellent
**Rating:** 8
**Confidence:** 4

**Summary:**

The study focuses on the development of unsupervised world models to enhance an autonomous agent's understanding of its environment. Though world models are a form of sequence modeling, their adoption in robotic applications like autonomous driving hasn't scaled as rapidly as language models such as GPT. Two primary challenges identified are the complex nature of observation spaces and the need for a scalable generative model. To address these, the research proposes a novel approach: (1) Tokenization of Sensor Observations. (2) Prediction using Discrete Diffusion. Applying this method to point cloud observations (which play a crucial role in autonomous driving) showed a significant improvement. The proposed model reduced the Chamfer distance by more than 65% for a 1-second prediction and over 50% for a 3-second prediction on major datasets like NuScenes, KITTI Odometry, and Argoverse2.

**Strengths:**

1. The contribution is clear and important to the autonomous driving society. Developing a driving world model is recognized as a critical step for scene understanding and decision-making.

2. This paper is well-written and easy to follow. The figures are intuitive and informative.

3. The experimental results are surprisingly good, which improves a lot over existing SOTA methods.

**Weaknesses:**

1. The metrics for evaluating the performance of the world model may not be reasonable enough. For point cloud, most of the points describe the background, which is usually static and irrelevant to the downstream task. The prediction of the motion of dynamic objects is more important. Maybe the author can also report the comparison results for dynamic objects or show the advantage of using such a model for some downstream tasks.

2. The conclusion says “One particularly exciting aspect of our approach is that it is broadly applicable to many domains. We hope that future work will combine our world modeling approach with model-based reinforcement learning to improve the decision-making capabilities of autonomous agents.”  It is unclear to me what the advantage of using such a world model is for MBRL, for example, compared to object segmentation and tracking pipelines.

3. There are many complex modules in the pipeline, including VQ-VAE, diffusion model, neural feature grid, and transformer. Not sure if it is easy to reproduce the results and extend it to other datasets or tasks.

**Questions:**

1. Could the authors elaborate more on “using past agent history as CFG conditioning improves world modeling”. What agent history is used here and how is it used? Does it introduce additional knowledge and make the comparison unfair?

2. Since the proposed method introduces a world model, could the authors demonstrate some qualitative examples of different future predictions with different actions? A longer horizon could be helpful to check the consistency of generated frames. I wonder how diverse the future prediction is and how accurate the prediction matches the given action.

---

> ### Author Response · Authors · 2023-11-19
> **Author Response**
>
> We thank the reviewer for their feedback. We are glad that the reviewer finds the paper’s contribution to be clear and important, and the writing to be easy-to-follow. We will be addressing your questions and concerns below.
>
> *Q: “The metrics for evaluating the performance of the world model may not be reasonable enough.”*
>
> A: The metrics we use follow the evaluation protocol from previous point cloud forecasting literature [1]. In addition, our qualitative results clearly show that our model drastically outperforms the previous SOTA on dynamic objects. See the second scene in Figure 5, the third scene in Figure 11, the second and third scene in Figure 15; those examples clearly show that our method can predict the motion of dynamic objects well even in complex scenes, while prior methods are unable to. In fact, our approach seems to be the first point-cloud forecasting method that can make reasonable predictions for dynamic objects without any labels.
>
> *Q: “It is unclear to me what the advantage of using such a world model is for MBRL, for example, compared to object segmentation and tracking pipelines.”*
>
> A: The main advantage is that it is an unsupervised learning algorithm, so that it can learn from any unlabeled data. The traditional object segmentation and tracking pipelines require annotations and labels to learn, which are very expensive.
>
> *Q: “There are many complex modules in the pipeline, including VQ-VAE, diffusion model, neural feature grid, and transformer. Not sure if it is easy to reproduce the results and extend it to other datasets or tasks.”*
>
> A: We have included all the technical details in the appendix for reproducibility. We have also added two additional diagrams, Figure 6 and Figure 7, about the architecture of VQ-VAE and spatio-temporal Transformer. It is important to recognize that our approach is conceptually simple: turning sensor observations into discrete tokens, and then applying discrete diffusion. It only has two models: the tokenizer and the world model. The complex designs are about specific architecture choices inside those two models: the neural feature grid is an architecture choice for the tokenizer decoder, and the transformer is an architecture choice for the world model. To extend the approach to other datasets and tasks (beyond self-driving), the specific architectural choices might need to change, but the general approach of tokenization and discrete diffusion are broadly applicable.
>
> *Q: “Could the authors elaborate more on ‘using past agent history as CFG conditioning improves world modeling’. What agent history is used here and how is it used? Does it introduce additional knowledge and make the comparison unfair?”*
>
> A: To better illustrate how CFG is applied in our world model, we have included an additional diagram Figure 8 in the Appendix. Section 4.3 also covers this part in detail. The agent history means the past observations and action history of the ego vehicle. We assigned a specific symbol to denote agent history: $c^{t-1}$. It is used as classifier-free diffusion guidance. The past observations and action history are already inputs in the world model; they do not require any additional knowledge.
>
> *Q: “Since the proposed method introduces a world model, could the authors demonstrate some qualitative examples of different future predictions with different actions?”*
>
> A: In our updated draft, we have included visualizations of predicted futures under counterfactual actions in Figure 10 (in the appendix). Here we modify the future trajectories of the ego vehicle (which are action inputs to the world model), and we demonstrate that the world model is able to imagine alternative futures given different actions. The imagined futures are consistent with the counterfactual action inputs.
>
> *References:*
>
> [1] Khurana, Tarasha, et al. "Point Cloud Forecasting as a Proxy for 4D Occupancy Forecasting." Proceedings of the IEEE/CVF Conference on Computer Vision and Pattern Recognition. 2023.

---

> > ### Comment · Reviewer_amws · 2023-11-20
> > **Reply to authors**
> >
> > Thanks to the authors for providing more explanation and experimental results. All my concerns are resolved so I am glad to raise my score.

---

### Official Review · Reviewer_2kSt · 2023-11-01

**Soundness:** 3 good
**Presentation:** 2 fair
**Contribution:** 3 good
**Rating:** 6
**Confidence:** 4

**Summary:**

The paper proposes to use a VQ-VAE+diffusion approach (similar to certain image diffusion pipelines) for the task of learning point-cloud world models for autonomous driving. They design task-specific encoder and decoder architectures to encode point-cloud observations as a sequence of discrete tokens, apply an interleaved spatial-temporal transformer and a discrete diffusion model to predict discrete codes for future frames, and decode with a model based on neural occupancy representations. The proposed model compares favorably to SOTA baselines for the task on standard metrics.

**Strengths:**

The proposed approach gives strong empirical performance. The architecture takes advantage of structure in the problem at several points in useful and interesting ways (in particular the combination of localized neural occupancy and BEV tokenization is quite interesting, and seems novel). I think the backbone (transformer+discrete diffusion) is comparatively less novel, but this is the first time I've seen it applied to the autonomous driving setting and it is interesting to see that it still gives strong performance.

Separately the authors propose several improvements to MaskGIT. These modifications seem to be crucial for the performance of their algorithm, but it would be interesting to see if these improvements generalize to the original image-based setting or to other discrete diffusion settings (though doing this on anything other than a toy problem would probably be outside of the scope of this paper).

The authors also identify an issue with standard point-cloud prediction metrics and propose a simple modification. Though this part may be less relevant to the ICLR community, it is an important observation for the self-driving community and should not be overlooked.

**Weaknesses:**

The proposed architecture is highly specific to point-cloud occupancy prediction (the novelty lies largely in the encoder and decoder, which are task-specific architectures).

The introduction/related work are somewhat intermixed, which is fine, but leads to confusing presentation. It's not entirely clear from the introduction/related work how the proposed method relates to MaskGIT, and although MaskGIT is heavily referenced it is never clearly described. Background of discrete diffusion could also be better described.

Ablations of differences to MaskGIT are good, but it would be useful to also present ablations of the other task-specific model components (encoder/decoder, and maybe also the BEV token grouping?) as compared to general-purpose versions.

Minor note: point cloud prediction visualizations are a little difficult to parse - I'm not sure how they could be improved but it's quite difficult to analyze results/see what's changed between two different images, both in Fig. 1 and Fig. 5.

**Questions:**

- It would be interesting to conduct a more thorough analysis of the modifications to MaskGIT, perhaps on a simpler discrete diffusion problem.
 - Why is L1 median reported inside the ROI but L1 mean reported for the full scene?
 - How important are the novel task-specific encoder and decoder (including the rendering and reconstruction losses on the encoder) to the final performance?
 - It's mentioned that the model is relatively small; how much do the results change when scaling the model (up/down)?

---

> ### Author Response · Authors · 2023-11-19
> **Author Response (Part 1)**
>
> We thank the reviewer for their feedback. We are glad that the reviewer finds our proposed method to be interesting and novel, and that the strong empirical performance of our method is recognized. We will be addressing your questions and concerns below.
>
> *Q: “The proposed architecture is highly specific to point-cloud occupancy prediction (the novelty lies largely in the encoder and decoder, which are task-specific architectures)”*
>
> A: It is true that our tokenizer is highly specific to point clouds. After tokenization, however, the specific 3D representations are abstracted away from the world model. The architecture of the world model is very general-purpose: it is a spatio-temporal Transformer, and both its inputs and outputs are discrete tokens.
>
> Additionally, the concept of Bird-Eye View (BEV) is quite general and goes beyond point clouds, since many popular approaches for camera-based perception such as Lift-Splat-Shoot [1] and BEVFormer [2] also lift camera data to BEV. Therefore, camera data can potentially be tokenized using BEV tokens as well. In this case, the spatio-temporal transformer acting as the world model can remain unchanged.
>
> *Q: “It's not entirely clear from the introduction/related work how the proposed method relates to MaskGIT, and although MaskGIT is heavily referenced it is never clearly described.”*
>
> A: In our updated draft, we have included a clear and concise description of MaskGIT in the background section. To include it here: “MaskGIT (Chang et al., 2022) has significantly simpler training and sampling procedures based on BERT alone: for training, it masks a part of the input tokens (with an aggressive masking schedule) and then predicts the masked tokens from the rest; for sampling, it iteratively decodes tokens in parallel based on predicted confidence.”
>
> *Q: “It would be useful to also present ablations of the other task-specific model components (encoder/decoder, and maybe also the BEV token grouping?) as compared to general-purpose versions”*
>
> A: Due to the cost and compute time required to train a tokenizer from scratch, it is too costly to ablate all components in the tokenizer. That being said, the individual components in the tokenizer are mostly off-the-shelf: the encoder follows standard designs in point-cloud based object detection; the vector quantization layer follows VQVAE; the decoder follows NeRF-like differentiable depth rendering already known to work well. Each off-the-shelf component was designed for general-purpose use; the contribution of our tokenizer is to demonstrate that combining the three components together can allow us to tokenize an entire self-driving scene with detailed reconstructions.
>
> *Q: “I'm not sure how they could be improved but it's quite difficult to analyze results/see what's changed between two different images, both in Fig. 1 and Fig. 5.”*
>
> A: In our updated draft, we have modified Figure 1 to provide a more zoomed-in view. We have also updated Figure 5 to provide zoomed-in orange boxes, which highlight that our method is significantly better in terms of not just the overall structure of the scene but also geometric details, especially at 3 seconds into the future. We will continue to find better ways to visualize the point clouds for the camera-ready version.
>
> *Q: “It would be interesting to conduct a more thorough analysis of the modifications to MaskGIT, perhaps on a simpler discrete diffusion problem.”*
>
> A: We agree with the reviewer that this would be interesting, but the focus of this work is unsupervised world modeling in large-scale self-driving scenes. Given that our experiment section is centered around point cloud forecasting, we think that perhaps it makes the most sense to leave such investigation to future work.
>
> *Q: “Why is L1 median reported inside the ROI but L1 mean reported for the full scene?”*
>
> A: As explained in the paper, L1 median within the ROI is a much better metric because (1) the median is more robust to outliers than the mean, and (2) in most self-driving applications predictions within the ROI is what matters for downstream tasks such as planning. For a more direct comparison with the baselines we still computed L1 mean since that is what was reported in their original work.

---

> > ### Author Response · Authors · 2023-11-19
> > **Author Response (Part 2)**
> >
> > *Q: “How important are the novel task-specific encoder and decoder (including the rendering and reconstruction losses on the encoder) to the final performance?”*
> >
> > A: To further clarify this issue, we want to note that the purpose of the tokenizer is to reconstruct the input, and that all components in the tokenizer are trained end-to-end from scratch. For the decoder, the rendering loss is necessary, since we use differentiable depth rendering to output the point clouds; the coarse reconstruction branch is also functionally necessary for spatial skipping to be used in differentiable rendering. Spatial skipping is already known to considerably speed up NeRF-like rendering [3], and is taken off-the-shelf. Regarding the encoder, its design follows standard point-based object detection backbones.
> >
> > *Q: “It's mentioned that the model is relatively small; how much do the results change when scaling the model (up/down)?”*
> >
> > A: While there are ways to further scale our Transformer world model, studying the scaling properties and scaling laws are beyond the scope of this paper, as it requires additional dedicated time and resources than what is feasible during this rebuttal period.
> >
> > *References:*
> >
> > [1] Philion, Jonah, and Sanja Fidler. "Lift, splat, shoot: Encoding images from arbitrary camera rigs by implicitly unprojecting to 3d." Computer Vision–ECCV 2020: 16th European Conference, Glasgow, UK, August 23–28, 2020, Proceedings, Part XIV 16. Springer International Publishing, 2020.
> >
> > [2] Li, Zhiqi, et al. "Bevformer: Learning bird’s-eye-view representation from multi-camera images via spatiotemporal transformers." European conference on computer vision. Cham: Springer Nature Switzerland, 2022.
> >
> > [3] Li, Ruilong, et al. "Nerfacc: Efficient sampling accelerates nerfs." arXiv preprint arXiv:2305.04966 (2023).

---

> > > ### Author Response · Authors · 2023-11-22
> > > **Author Response (Follow-up)**
> > >
> > > Since our initial response, we have managed to add ablation studies for the tokenizer as the reviewer suggested in the Appendix A.1.
> > >
> > >  - Table 7 provides an ablation on the effect of spatial skipping [1] (we have also added Figure 8 to better illustrate how spatial skipping works), where we train another tokenizer from scratch without the coarse reconstruction branch or the spatial skipping process; this tokenizer is only supervised with the depth rendering loss. In other words, this experiment studies what happens when we only use rendering loss without the (coarse) reconstruction loss. The table shows that spatial skipping improves point cloud reconstructions.
> > >  - We also show that only doing coarse reconstruction is not enough; differentiable depth rendering using implicit representation is crucial. In Figure 9, we provide a qualitative comparison between the VQVAE in UltraLiDAR [2] and our proposed tokenizer. The UltraLiDAR model only predicts whether a voxel has points present, and is similar to our coarse reconstruction branch. The results show that UltraLiDAR is unable to reconstruct fine-grained geometry due to the limited resolution of voxel predictions, whereas our tokenizer is able to produce high-fidelity point clouds that recover the details in the input point clouds. Figure 9 clearly shows that the results are qualitatively different, and using differentiable rendering is significantly better.
> > >
> > > We hope that those results answer the reviewer's question about the importance of the rendering and reconstruction losses, and we look forward to hearing from the reviewer.
> > >
> > > *References:*
> > >
> > > [1] Li, Ruilong, et al. "Nerfacc: Efficient sampling accelerates nerfs." arXiv preprint arXiv:2305.04966 (2023).
> > >
> > > [2] Xiong, Yuwen, et al. "Learning Compact Representations for LiDAR Completion and Generation." Proceedings of the IEEE/CVF Conference on Computer Vision and Pattern Recognition. 2023.

---

### Official Review · Reviewer_9NMX · 2023-11-03

**Soundness:** 4 excellent
**Presentation:** 4 excellent
**Contribution:** 4 excellent
**Rating:** 10
**Confidence:** 4

**Summary:**

The paper presents a state-of-the-art world model for driving data. Among several major contributions, the paper describes a new way to tokenize point clouds using a VQVAE combined with a PointNet; the paper also proposes a combination of generative masked modeling and discrete diffusion for learning a world model. The proposed method is tested on three commonly used lidar datasets and is shown to achieve state-of-the-art on 1s and 3s time horizon prediction.

**Strengths:**

1. The paper proposes a tokenizer for point clouds, which could have major applications across robotics.
2. The combination of MaskGIT with discrete diffusion and classifier-free guidance is novel. The idea of both decoding and denoising tokens is very interesting.
3. The proposed model outperforms prior state-of-the-art by a large margin.
4. The methods section is clear even though it proposes several novel models and losses.

**Weaknesses:**

The unnumbered first equation in Section 3 should be explained better.

Minor:
* It is not fully clear to me what “SE(3) ego poses” mean.
* Figure 1 and 5 might be easier to read if you zoom in on the circled areas.

**Questions:**

1. “We hope that future work will combine our world modeling approach with model-based reinforcement learning to improve the decision making capabilities of autonomous agents.” – Are you planning to release your code?
2. What hardware is required to train your model?

---

> ### Author Response · Authors · 2023-11-19
> **Author Response**
>
> We thank the reviewer for their feedback. We are glad that the novelty of our method and the significance of our results are recognized, and that the reviewer finds our method section to be clear. We will be addressing your questions and concerns below.
>
> *Q: “The unnumbered first equation in Section 3 should be explained better”*
>
> A: We have added additional explanations in the updated draft. This equation is the commonly used diffusion objective (Equation 13 in the original diffusion paper [1]).
>
> *Q: “It is not fully clear to me what ‘SE(3) ego poses’ mean”*
>
> A: We express the ego vehicle actions as ego pose transformations of the ego vehicle during two consecutive time steps, where each action is a 3-D homogeneous transformation matrix consisting of a translation and rotation (i.e., SE(3)). Since the world model is action-conditioned, we need to tell the world model where the agent plans to drive. As the ego vehicle can be assumed to have a rigid body, we do so in the form of rigid transformations. As mentioned in Section 4.4, we simply flatten those 4x4 matrices and feed the 16-dim vector as additional inputs to the spatio-temporal Transformer.
>
> *Q: “Figure 1 and 5 might be easier to read if you zoom in on the circled areas.”*
>
> A: In our updated draft, we have modified Figure 1 to provide a more zoomed-in view and better viewing angles. We have also updated Figure 5 to provide zoomed-in orange boxes, which highlight that our method is significantly better in terms of not just the overall structure of the scene but also geometric details.
>
> *Q: “Are you planning to release your code?”*
>
> A:  We are unable to release the code, but we have included all the technical details of our model in the appendix. We have also added two additional diagrams, Figure 6 and Figure 7, about our model architectures to improve reproducibility.
>
> *Q: “What hardware is required to train your model?”*
>
> A: The tokenizer is trained on 16 T4 GPUs; and the world model is trained on 8 A10 GPUs.
>
> *References:*
>
> [1] Sohl-Dickstein, Jascha, et al. "Deep unsupervised learning using nonequilibrium thermodynamics." International conference on machine learning. PMLR, 2015.

---

> > ### Comment · Reviewer_9NMX · 2023-11-22
> > **Response**
> >
> > Thank you for answering my question. I am in favor of accepting this paper.

---

### Official Review · Reviewer_2hr7 · 2023-11-06

**Soundness:** 3 good
**Presentation:** 4 excellent
**Contribution:** 3 good
**Rating:** 6
**Confidence:** 3

**Summary:**

The work presents a groundbreaking technique for learning world models in an unsupervised manner, with a particular application to autonomous driving. It addresses the complexity of interpreting unstructured sensor data by implementing a Vector Quantized Variational AutoEncoder (VQ-VAE) to tokenize this data, followed by the prediction of future states through a discrete diffusion process. This technique modifies the Masked Generative Image Transformer (MaskGIT) into a discrete diffusion model, which leads to a substantial increase in prediction accuracy. The proposed approach stands out for its ability to tokenize sensor inputs and utilize a spatio-temporal Transformer for the efficient decoding of future states, which has demonstrated an improvement in prediction accuracy over existing methods on autonomous driving datasets. The model achieves a significant reduction in prediction errors and also shows competence in generating both precise short-term forecasts and diverse long-term predictions, thereby holding great promise for the application of GPT-like learning paradigms in robotics.

**Strengths:**

The paper introduces a novel approach by combining VQ-VAE tokenization with a discrete diffusion process, which is kind of innovating. The idea of simplifying the observation space and tokenizing the observation space makes it much easier to model the complex observation space that are usually the case for self-driving.The proposed method's improvement is demonstrated through rigorous experimental validation, showing significant improvements in prediction accuracy over existing methods.The reduction in Chamfer distance for both short-term and long-term predictions indicates a high-quality advancement in the field of point cloud predictions. The paper is also well-structured, with a clear exposition of the methodology, which includes tokenization of sensor data and the subsequent prediction process.

**Weaknesses:**

The paper mostly addresses the prediction of near term future states but it is not clear if we go much further, how would the accuracy be? With a diffusion model, the inference could be slow, so this model may not be suitable for use on board but mostly would be useful for simulations and other tasks that don't require real time feedback or predictions. This may limit the application of this approach.

**Questions:**

How is the result if we predict much further, like 9s? In other dataset, like WOMD, the prediction horizon tends to be slightly longer so it would be great to know if the performance would drop significantly if we predict much further away states. Secondly, could you also share some insights on how this model could exactly be integrated within modern self-driving systems and work with other modules such as planning? How would noise in perception, like Lidar affect the prediction accuracy?

---

> ### Author Response · Authors · 2023-11-19
> **Author Response (Part 1)**
>
> Author Response:
>
> We thank the reviewer for their feedback. We are glad that the novelty of our approach and the rigor of our experimental evaluation are recognized, and that the reviewer finds our exposition to be clear and well-structured. We address the questions and concerns below.
>
> *Q: “The paper mostly addresses the prediction of near term future states but it is not clear if we go much further, how would the accuracy be?” “How is the result if we predict much further, like 9s?”*
>
> A: We answer this question from several perspectives:
>  - While we agree with the reviewer that long horizon prediction is an interesting problem, we believe finding a good metric to evaluate such capability will require extensive research. Evaluating sensor predictions 9s ahead via reconstruction metrics is not very meaningful. In autonomous driving scenarios, vehicles often cover a considerable distance within 9 seconds, leading to substantial transformations in the scene. Many foreground actors (like vehicles) and background elements (like trees) present in the new scene might not have been visible just 9 seconds earlier. Consequently, observations from 9 seconds prior often become insufficient, if not almost irrelevant, for making accurate future predictions. As such, unsupervised world modeling has a significantly greater degree of uncertainty than supervised object-level trajectory prediction, especially for long-horizon prediction. Note that supervised object-level trajectory prediction systems circumvent this issue by focusing solely on the trajectories of existing foreground actors that are visible at the current scene.
>  - Generating coherent long-horizon futures will likely require training the model to predict longer than 3s. While our framework allows us to train a model to predict 9s ahead, this will require significantly more compute resources, and is beyond the scope of our work.
>  - We note that our approach already significantly outperforms existing SOTA in 1s and 3s predictions, which have been the common benchmarks for unsupervised point cloud forecasting. Predicting 9s is very challenging, and there are no baselines to compare to, but we will consider it for future work.
>
> *Q: “with a diffusion model, the inference could be slow”*
>
> A: It is true that our current model requires 10 diffusion steps for generating each frame. However, we also point out the following:
>  - Inference speed is not our current focus in this work. The goal of our work is to showcase what results can be obtained with a discrete diffusion world model. As a pioneering work studying how to apply diffusion to unsupervised world modeling on real-world data, our model can already achieve good results using a reasonably small number of diffusion steps (10 steps).
>  - The field is actively working on improving the speed of diffusion models. Until recently, the field has primarily focused on Gaussian diffusion models such as DDPM [1], which initially required 1000 diffusion steps per sample; later on, DDIM [2] required 50 steps, progressive distillation [3] required around 8 steps, and Consistency Models [4] only required 1-2 diffusion steps. Even though discrete diffusion has not yet received as much attention from the field, we expect that similar distillation techniques can likely be applied in the future.
>
>
> *Q: “could you also share some insights on how this model could exactly be integrated within modern self-driving systems and work with other modules such as planning”*
>
> A: With a world model, the problem of autonomy can be seen as a model-based reinforcement learning (MBRL) problem. The autonomy system is an RL agent taking actions in the world. Previously, to apply MBRL, the bottleneck was about whether such a world model can be learnt on real-world observations in the self-driving scenes. Our work is a contribution towards removing this bottleneck. With an effective world model at hand, research can start to apply MBRL approaches such as Dreamer-v2 [5] to both learn a driving policy during training and plan with model-predictive control (MPC) at test time. While Dreamer-v2 assumes a given reward function, we can learn such a reward function from real-world driving data via techniques such as GAIL [6]. This is just one example of how this model can be integrated with modern autonomy systems.

---

> > ### Author Response · Authors · 2023-11-19
> > **Author Response (Part 2)**
> >
> > *Q: “How would noise in perception, like Lidar affect the prediction accuracy?”*
> >
> > A: All our experiments were conducted on real-world data (rather than simulated data), so our model can handle perception noise that naturally appears in real-world data. While studying the effect of perception noise is not the focus of our work, we highlight that the model also achieves a new SOTA for zero-shot transfer performance across datasets, where the model needs to transfer from one sensor type to another, as shown in Appendix A.1.
> >
> > *References:*
> >
> > [1] Ho, Jonathan, Ajay Jain, and Pieter Abbeel. "Denoising diffusion probabilistic models." Advances in neural information processing systems 33 (2020): 6840-6851.
> >
> > [2] Song, Jiaming, Chenlin Meng, and Stefano Ermon. "Denoising diffusion implicit models." arXiv preprint arXiv:2010.02502 (2020).
> >
> > [3] Salimans, Tim, and Jonathan Ho. "Progressive distillation for fast sampling of diffusion models." arXiv preprint arXiv:2202.00512 (2022).
> >
> > [4] Song, Yang, et al. "Consistency Models." arXiv preprint arXiv:2303.01469 (2023).
> >
> > [5] Hafner, Danijar, et al. "Mastering atari with discrete world models." arXiv preprint arXiv:2010.02193 (2020).
> >
> > [6] Ho, Jonathan, and Stefano Ermon. "Generative adversarial imitation learning." Advances in neural information processing systems 29 (2016).

---

> ### Comment · Reviewer_2hr7 · 2023-11-23
> **thanks for the rebuttal, I'll keep my rating.**
>
> Thanks for clarifying some of my questions. I think the notion of using a discrete diffusion process is indeed novel and improves the prediction accuracy nontrivially, so I keep my rating. But predicting 9s is still meaningful, though the author claimed they don't have the computing resources to do that which is reasonable. However, that does not mean that predicting 9s or some horizon longer than 3s isn't important. Since the vehicle speed could vary, in some cases, horizon longer than 3s is still important to predict to get a sense how far the vehicle is able to predict without making too much errors. There are indeed industry models that can predict long horizon trajectories, though some are not open sourced, which may make it hard to compare with.

---

### Meta-Review · Area_Chair_SKFq · 2023-12-06

**Metareview:**

This paper proposes a novel approach to world modelling and evaluates it on an established set of benchmark tasks for autonomous driving. The contribution of this paper was unanimously appreciated by all reviewers but most had outstanding questions that could be addressed in future work to further improve the contribution. Most notably, a very limited time horizon for future state predictions will often limit applicability of a world model.

**Justification For Why Not Higher Score:**

+ Highest rating reviewers left minimal feedback, remaining reviewers were closer to borderline
+ Limited time horizon of future predictions leave open questions on whether approach can scale.

**Justification For Why Not Lower Score:**

+ Unanimous positive support for acceptance from all reviewers
+ A novel model architecture evaluated on complex established benchmarks.

---

### Decision · Program_Chairs · 2024-01-16

Accept (poster)